# The anti-caspase 1 inhibitor VX-765 reduces immune activation, CD4+ T cell depletion, viral load, and total HIV-1 DNA in HIV-1 infected humanized mice

Mathieu Amand[1], Philipp Adams[1], Rafaela Schober[1], Gilles Iserentant[1], Jean-Yves Servais[1], Michel Moutschen[2], Carole Seguin-Devaux[1]*

[1]Department of Infection and Immunity, Luxembourg Institute of Health, Esch sur Alzette, Luxembourg; [2]Department of Infectious Diseases, University of Liège, CHU de Liège, Liège, Belgium

**Abstract** HIV-1 infection results in the activation of inflammasome that may facilitate viral spread and establishment of viral reservoirs. We evaluated the effects of the caspase-1 inhibitor VX-765 on HIV-1 infection in humanized NSG mice engrafted with human CD34+ hematopoietic stem cells. Expression of caspase-1, NLRP3, and IL-1β was increased in lymph nodes and bone marrow between day 1 and 3 after HIV-1 infection (mean fold change (FC) of 2.08, 3.23, and 6.05, p<0.001, respectively). IFI16 and AIM2 expression peaked at day 24 and coincides with increased IL-18 levels (6.89 vs 83.19 pg/ml, p=0.004), increased viral load and CD4+ T cells loss in blood (p<0.005 and p<0.0001, for the spleen respectively). Treatment with VX-765 significantly reduced TNF-α at day 11 (0.47 vs 2.2 pg/ml, p=0.045), IL-18 at day 22 (7.8 vs 23.2 pg/ml, p=0.04), CD4+ T cells (44.3% vs 36,7%, p=0.01), viral load (4.26 vs 4.89 log 10 copies/ml, p=0.027), and total HIV-1 DNA in the spleen (1 054 vs 2 889 copies /$10^6$ cells, p=0.029). We demonstrated that targeting inflammasome activation early after infection may represent a therapeutic strategy towards HIV cure to prevent CD4+ T cell depletion and reduce immune activation, viral load, and the HIV-1 reservoir formation.

*For correspondence:
carole.devaux@lih.lu

## Editor's evaluation

This important study examined the induction of inflammasome activation by HIV infection in a humanized NSG mouse model. The authors convincingly show that inflammasome activation plays a key role in CD4 T cell depletion and can be inhibited by the anti-caspase 1 inhibitor VX-765. The results are of interest to scientists and physicians interested in the treatment and pathogenesis of HIV-1 infection.

## Introduction

Around 38 million people worldwide are living with the human immunodeficiency virus (HIV) and every year, 1. 7million people get newly infected (*UNALDS, 2020*). Although combined antiretroviral therapy (cART) can suppress viremia and dramatically improved the life expectancy and the clinical outcome of HIV-1 infected patients (*Antiretroviral Therapy Cohort Collaboration, 2008*), optimal cART treatment is not curative due to the persistence of a lifelong HIV-1 replication competent reservoir (*Siliciano et al., 2003*). In addition, cART does not fully resolve HIV-associated immune abnormalities including low-grade chronic inflammation, immune activation/dysfunction, and poor CD4+ T cell reconstitution (*French et al., 2009*; *Hunt et al., 2003*) and imposes a high burden to the health care

**eLife digest** The human immunodeficiency virus (HIV) affects millions of people across the world, and has caused over forty million deaths. HIV attacks the immune system, eventually leading to lower levels of immune cells, which prevent the body from fighting infections. One of the early effects of HIV infection is inflammation, an immune process that helps the body remove foreign invaders like viruses. Unfortunately, long term inflammation can lead to serious conditions like cardiovascular disease and cancer.

Doctors manage HIV using a class of drugs known as antiretrovirals. These drugs reduce the amount of virus in the body, but they cannot eliminate it entirely. This is because, in the early days of infection, copies of the virus build up in certain organs and tissues, like the gut, forming viral reservoirs. Antiretroviral drugs cannot reach these reservoirs to eliminate them, making a cure for HIV out of reach. One way to address this problem is to develop a new class of drugs that can stop the virus from forming these reservoirs in the first place.

Amand et al. wanted to see whether they could reduce the amount of viral reservoirs that form in HIV patients by interrupting a process called inflammasome activation, which occurs early after HIV infection. Inflammasomes are viral detectors that play a role in both inflammation and the formation of viral reservoirs. They activate an enzyme called caspase-1, which in turn activates proteins called cytokines. These cytokines go on to stimulate further inflammation.

Amand et al. wanted to see whether a drug called VX-765, which blocks the activity of the caspase-1 enzyme, could reduce inflammation and stop the formation of viral reservoirs. To do this, Amand et al. first 'humanized' mice, by populating them with human immune cells, so they could become infected with HIV. They then infected these mice with HIV, and proceeded to treat them with VX-765 two days after infection. The results showed that these mice had fewer viral reservoirs, lower levels of cytokines and higher numbers of immune cells than untreated mice.

The findings of Amand et al. show that targeting inflammasome activation early after infection could be a promising strategy for treating HIV. Indeed, if similar results were obtained in humans, then this technique may be the road towards a cure for this virus. In any case, it is likely that combining drugs like VX765 with antiretrovirals will improve long term outcomes for people with HIV.

system due to treatment resistance, adverse events, and high treatment costs. Recent evidence indicates that inflammation drives HIV-1 reservoirs persistence which in turn contributes to inflammation, creating a pathogenic vicious cycle (*Klatt et al., 2013*). These inflammatory and immune disorders predispose HIV-1 infected patients under cART to non-AIDS age-related comorbid conditions such as cardiovascular diseases, cancer, and neurodegenerative diseases and are predictive of an increased risk of morbidity and mortality (*Kuller et al., 2008*). Strategies targeting the mechanisms of HIV-induced inflammation are now evaluated to reduce HIV-1 reservoirs, a fundamental requirement to achieve an HIV cure (*Massanella et al., 2016*).

HIV infection triggers immune activation/dysfunction and inflammation through several mechanisms such as direct effects of viral proteins on immune cells (*Rieckmann et al., 1991*; *Lee et al., 2003*; *Simmons et al., 2001*), persistent production of type I and II interferons driven by activation of toll-like receptors with HIV-1 RNA (*Sereti and Altfeld, 2016*), fibrosis of lymphoid structure linked to CD4$^+$ T cells depletion (*Beignon et al., 2005*) and mucosal barriers damages associated with microbial translocation (*Klatt and Brenchley, 2010*; *Gordon et al., 2010*). Although inflammation can contribute to the early control of HIV infection, immune dysfunction and chronic inflammation can in turn promote viral spread and the replenishment of the reservoir by stimulating the migration of CD4$^+$ T cells to sites of viral replication (*Klatt and Silvestri, 2012*; *Hunt et al., 2011*), generating new activated target CD4$^+$ T cells (*Biancotto et al., 2008*). This further leads to a reduced ability of the immune system to kill infected cells, and an increased cell-surface expression of immune checkpoints which contributes to the survival of infected cells (*Hatano et al., 2013*; *Chomont et al., 2009*). Furthermore, its stimulates latently infected cells to proliferate but also to produce the virus (*Chun et al., 1998*; *Cameron et al., 2010*). Altogether, inflammation, poor antiviral immune response, and HIV-1 persistence create a pathogenic cycle. Interfering with mechanisms linking inflammation and HIV-1 persistence could contribute to an HIV cure and the management of inflammation-linked

comorbidities. *Micci et al., 2015* established the proof of concept that targeting chronic inflammation with IL-21, a potent immunomodulatory cytokine, in ART-treated SIV-infected non-human primates could improve disease prognosis by reducing immune activation and the viral reservoir. To achieve HIV cure, as the viral reservoir is set within few days after infection (*Lewis, 2016*), early initiation of novel treatment is required to block reservoirs establishment and may protect the immune system (*Rutstein et al., 2017*).

Inflammasomes plays important roles in innate immunity and inflammation as multiprotein signalling platforms sensing pathogens, danger signals, and regulating cellular functions (*Broz and Dixit, 2016*). Upon recognition of the microorganism or danger signals, certain Pattern Recognition Receptors (PRRs) recognize the presence of unique microbial components, so-called pathogen-associated molecular patterns (PAMPs) or damage-associated molecular patterns (DAMPs), and recruit a multiprotein signalling platform containing the adapter protein ASC (apoptosis-associated speck-like protein containing a caspase activation and recruitment domain) which induces the proteolytic activation of the procaspase-1 (*Broz and Dixit, 2016*). Subsequently, active caspase-1 initiates the release of pro-inflammatory cytokines such as IL-1β and IL-18 and the induction of pyroptosis, an highly inflammatory form of cell death. HIV-1 was described to induce the formation and action of an inflammasome in several cell types and models (*Doitsh et al., 2014*; *Biancotto et al., 2007*; *Ahmad et al., 2002*; *Song et al., 2006*; *Wiercinska-Drapalo et al., 2004*; *Granowitz et al., 1995*; *Pugliese et al., 2002*; *Ahmad et al., 2018*; *Guo et al., 2014*; *Chattergoon et al., 2014*; *Pontillo et al., 2012*; *Pontillo et al., 2013*; *Walsh et al., 2014*; *Jakobsen et al., 2013*). Levels of IL-1β and IL-18 were found to be elevated in HIV-1 infected individuals (*Biancotto et al., 2007*; *Ahmad et al., 2002*; *Song et al., 2006*) and to induce HIV-1 production (*Wiercinska-Drapalo et al., 2004*; *Granowitz et al., 1995*; *Pugliese et al., 2002*). Ahmad et al. found that a higher proportion of blood monocytes from HIV-1 infected patients were ASC specks positive, a hallmark of inflammasome activation (*Ahmad et al., 2018*).

In innate immune cells, such as monocytes/macrophages (*Guo et al., 2014*; *Chattergoon et al., 2014*), dendritic cells (*Pontillo et al., 2012*; *Pontillo et al., 2013*) and microglial cells (*Walsh et al., 2014*), HIV-1 infection results in the activation of an inflammasome involving the PRR nucleotide-binding oligomerization domain (NOD), leucine-rich repeat (LRR)-containing proteins (NLR) family member 3 (NLRP3). Upon activation of NLRP3 inflammasome, pro-IL-1β is processed by activated caspase 1 to form active IL-1β. NLRP3 inflammasome activation requires a priming signal and an activation signal ultimately leading to pro-inflammatory cytokine release and pyroptosis. More specifically, in human monocytes/macrophages, HIV-1 triggers the NLRP3 inflammasome through the detection of HIV-1 RNA by TLRs and the activation of the NF-κB pathway (*Jakobsen et al., 2013*). Interferon (IFN)-inducible protein 16 (IFI16) and AIM2 are DNA sensor of the AIM2-like receptors (ALRs) family and can form an inflammasome with ASC upon detection of nuclear DNA. IFI16 is also able to sense HIV-1 in macrophages, but it is unknown whether it can activate an inflammasome in such cell type (*Jakobsen et al., 2013*). Using lymphoid aggregate culture (HLAC) ex vivo systems from fresh human tonsil or spleen tissues, Doitsh et al. revealed how most of CD4⁺ T cells dies during HIV-1 infection (*Doitsh et al., 2010*). Over 95% of CD4 T cells are non-permissive to HIV-1 infection and accumulates incomplete DNA transcripts in their cytosol. These transcripts are detected by IFI16, a sensor which activates caspase-1, triggers pyroptosis and the release of intracellular content and pro-inflammatory cytokines including IL-1β (*Monroe et al., 2014*). The role of the inflammasome in early SIV pathogenesis has recently been shown in the impairment of the innate and adaptive immune response (*Barouch et al., 2016*) thereby facilitating viral replication (*Lu et al., 2016*).

Altogether, these results strongly suggest that inflammasome activation occurs early during HIV-1 infection, and could be a central mechanism in the early viral escape to the immune system and furthermore, may facilitate the establishment of the pathogenic cycle of HIV-1 reservoirs persistence and inflammation. VX-765, a selective caspase 1 inhibitor, reduces activation and activity of caspase 1 and antagonizes NLRP3 inflammasome assembly and activation in the context of several inflammatory disorders and diseases such as atherosclerosis, sepsis, or alzheimer disease (*Jin et al., 2022*; *Wu et al., 2021*; *Flores et al., 2020*). Therefore, the aim of our study was to investigate the effects of the Caspase-1 inhibitor VX-765 in an humanized mouse model of HIV-1 infection as a therapeutic strategy to reduce HIV-1-induced inflammation and reservoirs establishment.

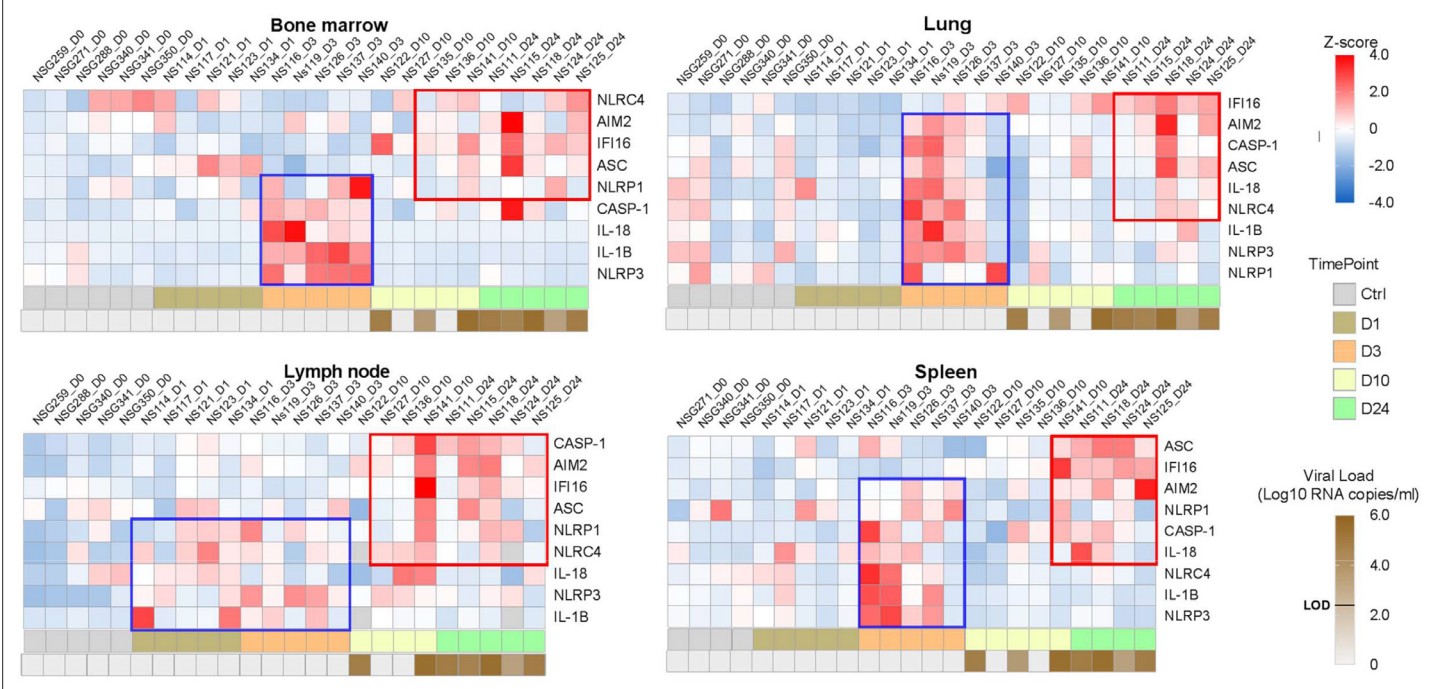

**Figure 1.** Heatmaps reveal a biphasic inflammasome-related genes transcriptomic upregulation upon HIV-1$_{JRCSF}$ infection of huNSG mice. Relative mRNA expression of the indicated genes measured by qPCR in hCD45$^+$ cells isolated from bone marrow, lung, lymph nodes and spleen in HIV-1$_{JRCF}$ i.p. infected huNSG mice at day 0 (n=6), day 1 (n=5), day 3 (n=5), day 10 (n=5), and day 24 (n=5). Gene expression was standardized using z-score method. Red and blue colors identify up- and down-regulated genes, respectively.

The online version of this article includes the following figure supplement(s) for figure 1:

**Figure supplement 1.** Kinetics of viremia and early CD4$^+$ T cell loss during HIV-1 infection of huNSG mice.

**Figure supplement 2.** HIV-1 disseminates within 3 days post-infection in huNSG mice.

**Figure supplement 3.** HIV-1 induction of inflammasome related genes expression in tissues.

## Results

### Early inflammasome-related genes expression upon HIV-1 infection

Humanized mice recapitulate key aspects of HIV-1 infection and pathogenesis (**Singh et al., 2012**; **Adams et al., 2021**). However, whether they recapitulate inflammasome activation upon HIV-1 infection remains undetermined. We first investigated whether inflammasome activation contributes to the early host response during HIV-1 infection in humanized mice (**Figure 1**). We generated human CD34$^+$ hematopoietic stem cells-engrafted NOD/SCID/IL2rγ$^{-null}$ (huNSG) mice and performed serial necropsies on day 0 (n=6), day 1 (n=5), day 3 (n=5), day 10 (n=5), and day 24 (n=5) post-intraperitoneal injection of 5×10$^4$ TCID$_{50}$ HIV-1$_{JRCSF}$. Viremia was detectable in 3 (60%) and 5 (100%) animals sacrificed on day 10 and 24 post-infection (p.i.), respectively (**Figure 1—figure supplement 1A**). Acute HIV-1 infection in huNSG mice was associated with a profound decrease in CD4$^+$ T cell percentages in the blood at day 24 when compared to animals sacrificed at day 10 post-infection (**Figure 1—figure supplement 1B**).

Tissue viral RNA levels in hCD45$^+$ cells from several tissues were further quantified by RT-PCR (**Figure 1—figure supplement 2**). All huNSG mice necropsied at day 1 following HIV-1$_{JRCSF}$ challenge were negative for viral RNA, whereas huNSG mice sacrificed at day 3 displayed at least one tissue positive for viral RNA with the spleen and the lung being positive in 80% (4 of 5) and 60% (3 of 5) animals, respectively, indicating viral dissemination at least within 72 hr. Tissue viral RNA levels became detectable in the lymph nodes (5 of 5) and the blood (4 of 5) in animal sacrificed on day 10 p.i. The number of viral RNA copies varied widely among animals sacrificed at day 7 and was elevated in most animals sacrificed at day 24.

Previous reports have highlighted increased transcripts levels of inflammasome components upon HIV-1 infection in HIV-1 infected patients (*Biancotto et al., 2007*), SIV-infected rhesus monkeys (*Barouch et al., 2016*) and in vitro models infection (*Doitsh et al., 2010*). To show evidences of early inflammasome sensing of HIV-1 in huNSG mice upon HIV-1 infection, we quantified mRNA expression of genes involved in inflammasome activation (*NLRP3, NLRP1, NLRC4, AIM2, IFI16, ASC, CASP1, IL1B, and IL18*) in hCD45+ cells from multiple tissues collected at necropsy. We observed a bi-phasic induction of inflammasome related-genes upon HIV-1 infection in the bone marrow, lymph nodes, and spleen (*Figure 1*). In the bone marrow, a striking but short-lasting early upregulation of inflammasome components appeared around 3 days p.i. These included the NOD-like receptor NLRP3 (p<0.05), the adaptor ASC (p<0.01), bridging NLRs such as NLRP3 to the inflammatory Caspase-1 (CASP-1) (p<0.01) and the downstream pro-inflammatory cytokines IL-1β (p<0.01)and IL-18 (p<0.01)indicating the up-regulation of the whole NLRP3 pathway. In the lymph nodes, the induction of an early inflammasome response was evident as soon as day 1 and 3 p.i. with the upregulation of the NOD-like receptors NLRP1 (p<0.05) and NLRP3 (p<0.01), AIM2 (p<0.05), CASP-1 (p<0.01) and IL-1b (p<0.01).

Interestingly, in contrast to the bone marrow, this initial burst was sustained in the lymph nodes with the prolonged upregulation of several genes (p≤0.05 for *NLRP3, AIM2*, and *CASP1*) up to day 24 p.i. In the spleen, *AIM2* was the only significantly upregulated gene at 3 days p.i. (p<0.05), although we observed a tendency for other genes (*NLRP3, NLRC4, CASP1*, and *IL1B*) (*Figure 1—figure supplement 3*). Altogether, these rapid inflammasome-related transcriptomic changes indicate an early sensing of HIV-1 infection, especially in the bone marrow and lymph nodes, before HIV-1 RNA was quantifiable in tissues (*Figure 1*) and may reflect the host response to the inoculum challenge virus.

## IFI16 and AIM2 expression correlates with HIV-1 replication and pathogenesis

IFI16 and AIM2 are DNA sensors of the ALRs family and can form an inflammasome with ASC upon detection of nuclear DNA (*Guo et al., 2015*). Importantly, IFI16 restricts HIV-1 infection in macrophages

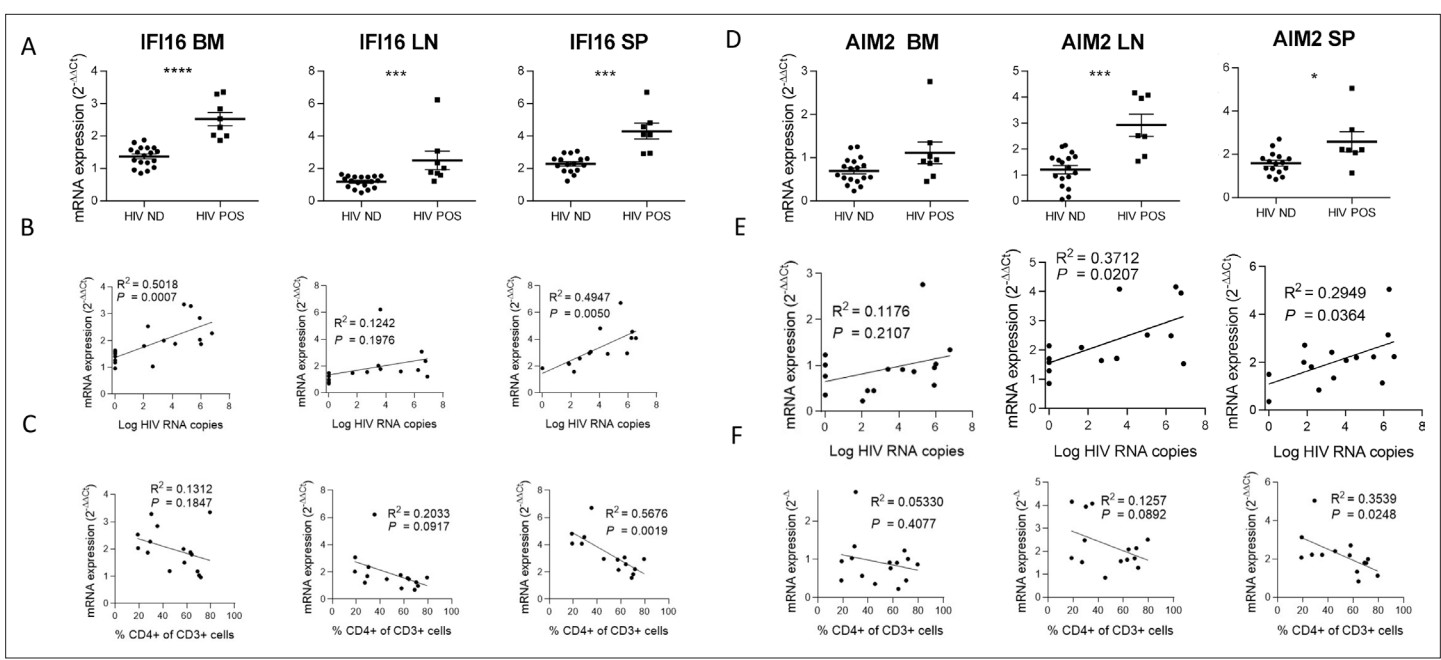

**Figure 2.** IFI16 and AIM2 expression correlates with HIV-1 replication and pathogenesis. IFI16 and AIM2 relative mRNA expression comparison between viremic (HIV POS, n=8) and aviremic (HIV ND, n=16) huNSG mice (**A and D**) in the bone marrow (BM), lymph nodes (LN), and the spleen (SP). Correlation between IFI16 or AIM2 relative mRNA expression with corresponding tissue viral RNA copies (**B and E**) or CD4 T cells percentage in the blood (**C and F**) for infected animals showing detectable HIV-1 mRNA in human CD45+ cells (n=15). qPCR results were analyzed using the 2^{-ΔΔCT} method. Statistical tests were performed by Mann–Whitney t-tests for comparison of two groups (* p<0.05, *** p<0.0005, **** p<0.0005) or Spearman correlation tests.

The online version of this article includes the following figure supplement(s) for figure 2:

**Figure supplement 1.** ASC and CASP-1 expression correlates with HIV-1 replication and pathogenesis.

but promotes HIV-1 induced CD4+ T cells death by pyroptosis and thus may drive HIV-1 pathogenesis (**Doitsh et al., 2010**; **Monroe et al., 2014**). In accordance, viremic mice demonstrated higher levels of IFI16 expression in the organs than in non-viremic mice (p<0.0001, 0.001, 0.001 in bone marrow, lymph nodes, and spleen) (**Figure 2**). Interestingly, in our humanized mice model, the upregulation of IFI16 was common in all three organs, with a maximal transcription at 24 days p.i. (p<0.01, 0.05 and 0.01 in bone marrow, lymph nodes, and spleen, respectively) (**Figure 1—figure supplement 3**), After the initial burst, delayed transcriptomic changes occurring around day 24 p.i. were observed, notably in the spleen. Furthermore, IFI16 expression was higher in viremic mice (**Figure 2A**) and correlated with corresponding tissue viral RNA copies (**Figure 2B**; p=0.0007 in bone marrow and p=0.005 in the spleen) and with CD4+ T cells percentage in the blood (**Figure 2C**; p=0.0019 for the spleen).

Similarly, AIM2 reached a maximum of expression at day 24 p.i. in the bone marrow, lymph nodes and the spleen (p<0.05, p<0.005 and p<0.005, respectively) (**Figure 1—figure supplement 3**) and was expressed at higher levels in viremic mice when compared to non-viremic mice (p<0.001 and 0.05 in lymph nodes and spleen, respectively) (**Figure 2D**). AIM2 expression also correlated with viral RNA copies in the same organ (**Figure 2E**; p<0.05 in lymph nodes and spleen) and inversely with CD4+ T cells percentage in the blood (**Figure 2F**; p<0.05 in the spleen). In addition, other inflammasome-related genes (*ASC* and *CASP1*) reached a maximum of expression 24 days p.i., notably in the spleen (p<0.01; *ASC, CASP1*) and in the lymph nodes (p<0.01; *CASP1*) (**Figure 2—figure supplement 1**) and their expression was associated with HIV-1 replication in the bone marrow for ASC (p<0.05) and HIV-1 pathogenesis for ASC in the bone marrow and the spleen (p<0.05) and for CASP-1 in the lymph nodes (**Figure 2—figure supplement 1**, p<0.05).

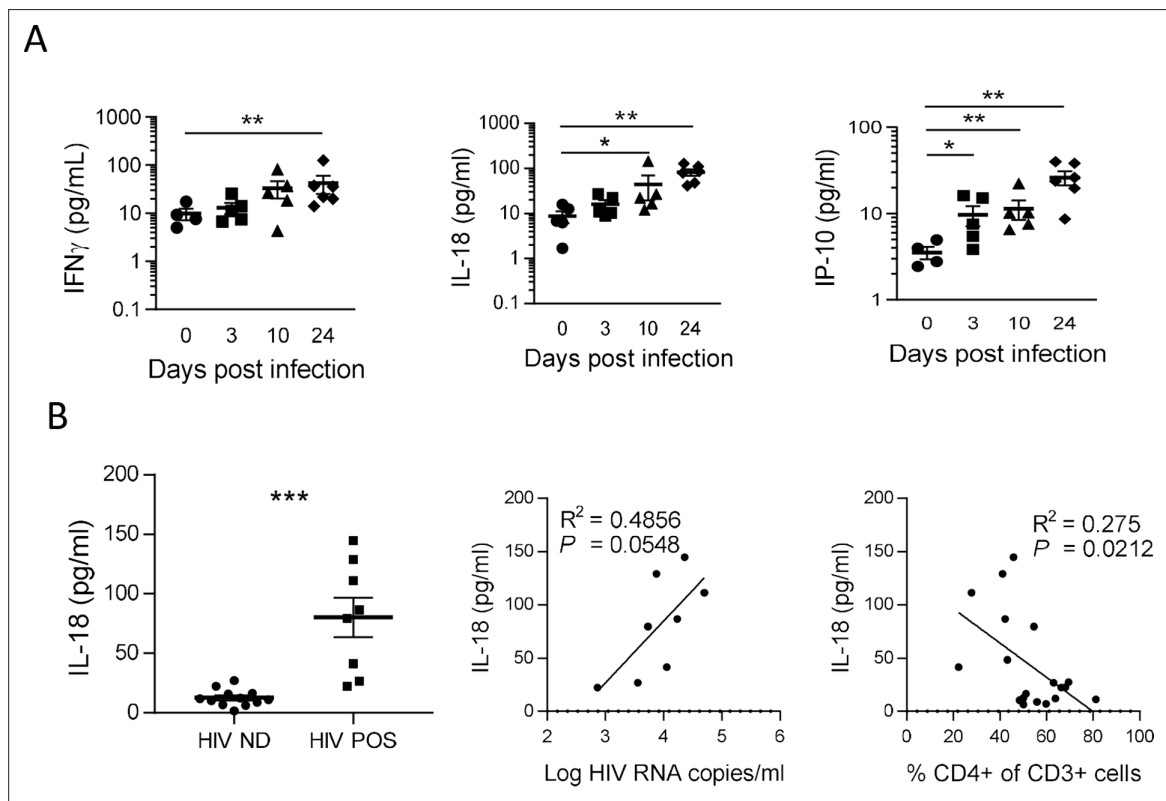

**Figure 3.** IL-18 is induced by HIV-1 replication and correlates inversely with CD4+ T cells percentage. (**A**) Plasma cytokines concentration were measured using a multiplex U-PLEX Biomarker assay kit in the plasma of HIV-1_JRCF i.p. infected huNSG mice at day 0 (n=5), day 3 (n=5), day 10 (n=5), and day 24 (n=5). (**B**) Plasma IL-18 levels comparison between viremic (HIV POS, n=8) and aviremic (HIV ND, n=12) huNSG mice. Correlations between plasma IL-18 levels with corresponding plasma viral load (**C**) or CD4 +T cells percentage in the blood (**D**). Statistical tests were performed by Mann–Whitney t-tests for comparison of two groups (* p<0.05, ** p<0.005) or Spearman correlation tests.

The online version of this article includes the following figure supplement(s) for figure 3:

**Figure supplement 1.** IFN-γ and IP-10 are inducted by HIV-1 replication.

## HIV-1 infection induces the release of inflammasome-related cytokines

Typical inflammasome activation requires a two-step process including (1) a priming signal resulting in the transcriptional upregulation of inflammasome-related genes and (2) a second signal causing inflammasome assembly and activation at the protein level. Upon assembly, the inflammasome induces the proteolytic activation of procaspase-1 into active caspase-1 that initiates the maturation and release of inflammasome-related pro-inflammatory cytokines such as IL-1β and IL-18 (*Guo et al., 2015*). Therefore, qPCR measurements of inflammasome-related gene expression mostly reflect the priming steps (*Figure 2*) while inflammasome activation usually requires the quantification of mature IL-1β and IL-18 or caspase-1 activity.

To confirm inflammasome activation in huNSG mice upon HIV-1 infection, we next quantified plasma levels of inflammasome-related (IL-1β and IL-18) and others pro-inflammatory cytokines (IFN-γ, IL-6, IP-10, and TNF-α), as shown in *Figure 3*. Plasma was collected at necropsy on day 0 (n=5), day 3 (n=5), day 10 (n=5), and day 24 (n=6) post-infection. HIV-1 infection was previously described to induce a rapid and intense cytokine storm involving several pro-inflammatory biomarkers (*Stacey et al., 2009*). Concordantly, we observed a rapid elevation in the plasma levels of several pro-inflammatory cytokines such as IFN-γ, IL-18, and IP-10 but not of IL-6 and TNFα that remained below the LOD in most of mice plasma upon HIV-1 infection (*Figure 3A* and *Figure 3B*). Interestingly, IL-18 levels accumulated at day 10 and day 24 post-infection (*Figure 3A*; p<0.05 and 0.01, respectively) and viremic mice harboured higher IL-18 levels than non-viremic mice (*Figure 3B*; p<0.001). Furthermore, their plasma levels correlated with viremia (*Figure 3B*; p=0.0548) and inversely with CD4+ T cells percentage in the blood (*Figure 3B*; p=0.0212). Although IFN-γ and IP-10 concentrations were also higher in viremic mice as compared to non-viremic mice, plasma levels were not correlated with plasma viral load or CD4+ T cells percentage in the blood (*Figure 3—figure supplement 1*). IL-1β levels were below the LOD of the assay for most of the mice blood samples. Its short half-life (*Kudo et al., 1990*), the low volume of plasma collected in mice, and its fast degradation might explain that that we did not catch

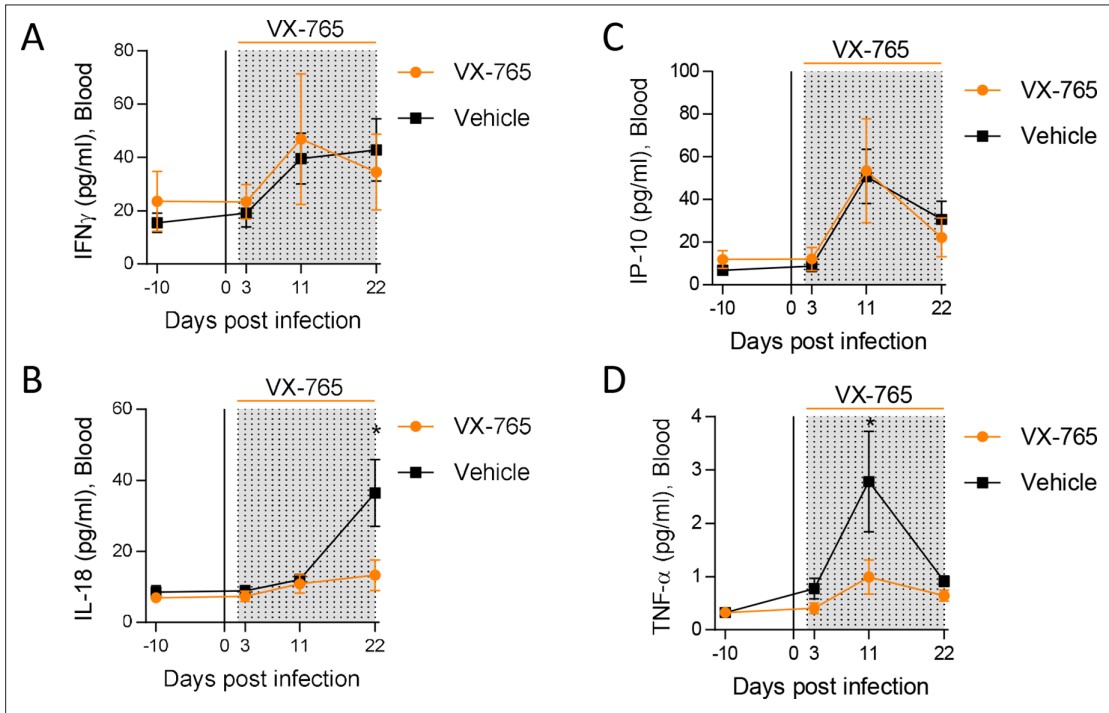

**Figure 4.** Caspase-1 inhibition reduces circulating levels of IL-18. Plasma cytokine concentrations were measured using a multiplex U-PLEX Biomarker assay kit in longitudinal plasma samples of HIV-1$_{JRCF}$ i.p. infected huNSG mice treated with VX-765 (VX-765, n=12) or vehicle (Vehicle, n=12) at 3 days, 11 days and 22 days after infection. Statistical tests were analysed by Mann–Whitney t-tests for comparison of two groups (* p<0.05).

The online version of this article includes the following figure supplement(s) for figure 4:

**Figure supplement 1.** Caspase-1 inhibition reduces levels of plasma cytokines and mRNA expression of inflamasomme genes.

**Figure supplement 2.** VX-765 treatment prevents CD4+ T cell depletion in the blood.

a significant early increase for this cytokine, active in the low ng/mL or even in the pg/mL range in wt mice. Altogether, these results strongly indicate that an inflammasome is activated upon HIV-1 infection in our humanized mice model and is associated with disease progression.

### Inflammasome inhibition reduces HIV-1-induced circulating levels of IL-18 and the cytokine storm in HIV-1 infected huNSG mice

To decipher the impact of inflammasome on HIV-1 pathogenesis in vivo, we treated huNSG mice daily by intraperitoneal injection of the caspase-1 inhibitor VX-765 (*Wannamaker et al., 2007*) at the dose of 200 mg/kg (*Zhang and Zheng, 2016*; *Ravizza et al., 2008*) for 21 consecutive days starting on day 2 after HIV-1 infection to cover the early upregulation of inflammasome-related genes. As expected, HIV-1 infection in the vehicle group induced a potent cytokine storm with elevated plasma levels of the inflammasome-related cytokines IL-18 but also IFN-γ, IP-10 and TNFα when compared to day 10 before infection (p<0.05; 0.01; 0.05; 0.001; 0.01; 0.001, respectively; *Figure 4* and *Figure 4—figure supplement 1*). In contrast, caspase-1 inhibition blunted the amplitude of the HIV-1 induced cytokine storm as VX-765 treated huNSG mice only upregulated significantly the levels of IP-10 and TNF-α after infection (p<0.05 and 0.05, respectively; *Figure 4—figure supplement 1E and F*). Importantly, caspase-1 inhibition reduced the circulating levels of the inflammasome-related cytokine IL-18 at day 22 in comparison with the vehicle group (p<0.05; *Figure 4B*). TNF-α levels at day 10 were also reduced under VX-765 treatment (p<0.05, *Figure 4D*) when compared to the vehicle group. As expected, at the level of inflamasomme gene expression, VX-765 decreased the expression of *IFI16*, *AIM2*, *ASC*, and *CASP1* genes, both at 11 and 22 days after HIV infection in the spleen as compared to vehicle-treated mice (*Figure 4—figure supplement 1*). Altogether, these results confirm that the inhibition of inflammasome upon HIV-1 infection in huNSG mice reduced the levels of circulating pro-inflammatory cytokines induced during early HIV-1 infection.

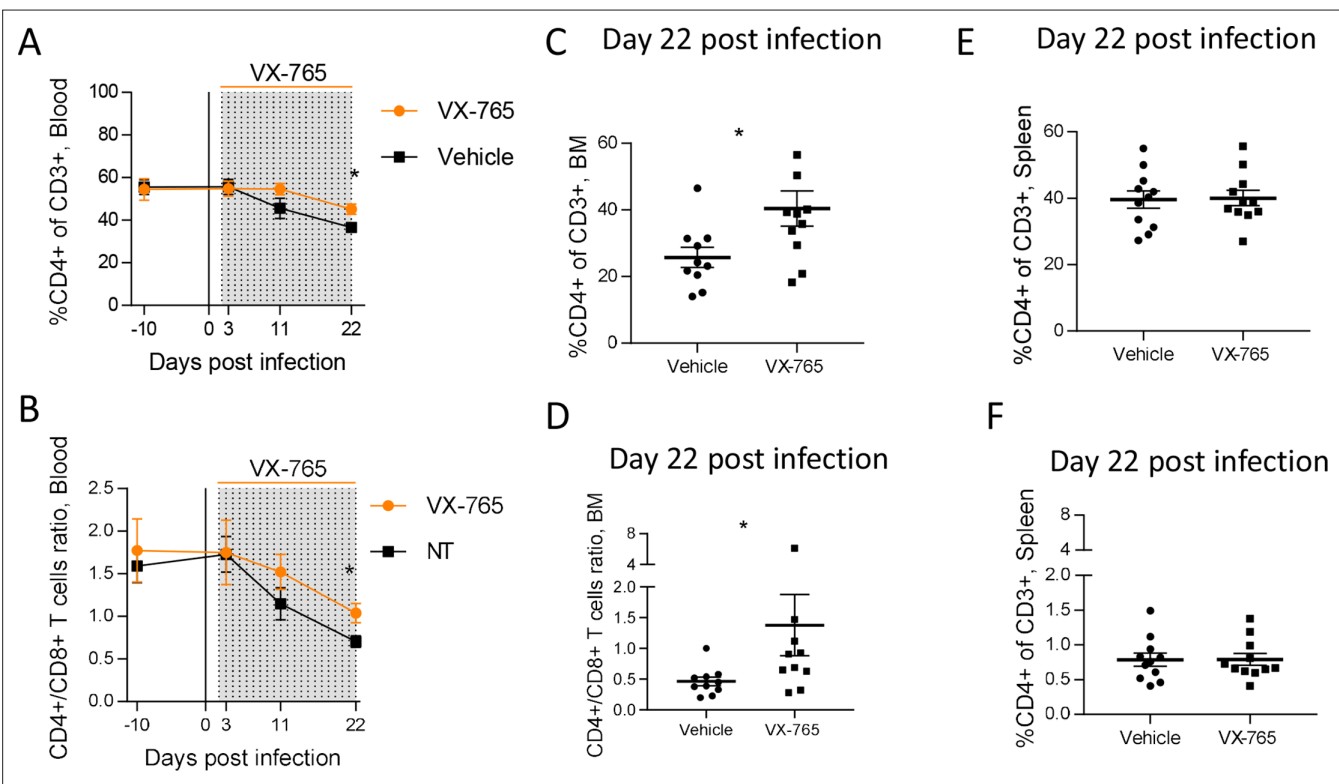

**Figure 5.** VX-765 treatment prevents CD4+ T cell depletion in the blood and bone marrow (BM) during HIV-1 infection of huNSG mice. Blood human CD4+ T cells percentage among CD3+ cells (**A**) and CD4+/CD8+ T cell ratios (**B**) measured longitudinally by flow cytometry in VX-765 (VX-765, n=12) or vehicle (Vehicle, n=12) huNSG mice at 3 days, 11 days, and 22 days after HIV-1 infection. Bone marrow (BM) (**C and D**) and spleen (**E and F**) human CD4+ T cells percentage among CD3+ cells and CD4+/CD8+ T cell ratios, respectively, measured at day 22 post-infection in vehicle or VX-765-treated HIV-1 infected huNSG mice. Statistical tests were analysed by Mann–Whitney t-tests for comparison of two groups (* p<0.05).

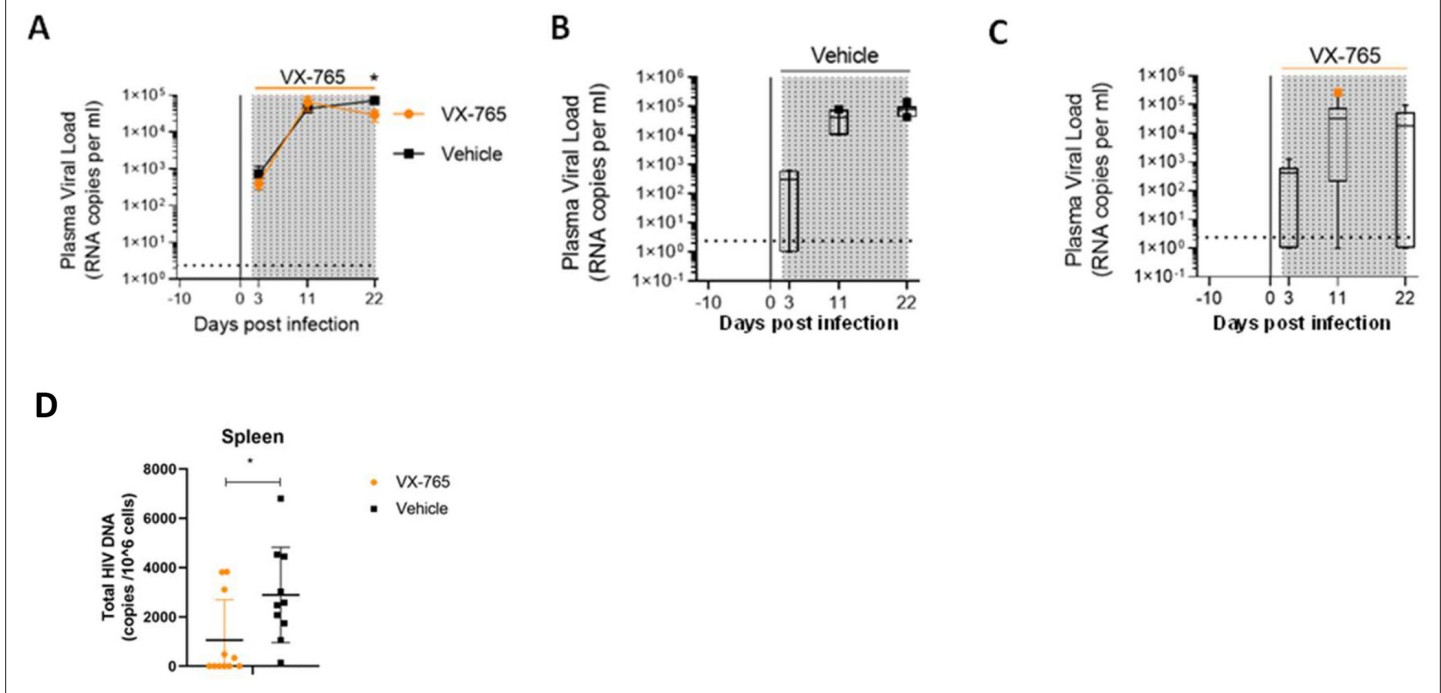

**Figure 6.** VX-765 treatment reduces plasma viral load and total HIV-1 DNA after HIV-1 infection of huNSG mice. Plasma viral load (**A**) was measured longitudinally by ddPCR in vehicle (Vehicle, n=12) (**B**) or in VX-765 (VX-765, n=12) (**C**) or in treated HIV-1 infected huNSG mice at 3 days, 11 days, and 22 days after HIV-1 infection. Total HIV-1 DNA was measured by qPCR at day 22 post-infection in the spleen of CD45+ human cells isolated by MACS purification from all HIV-1 infected huNSG mice treated (VX-765, n=12) or not (Vehicle, n=12) with VX-765 (**D**). Statistical tests were analysed by Mann–Whitney t-tests for comparison of two groups (* p<0.05).

## Administration of VX-765 prevents CD4⁺ T cell depletion and decreases viral load and total HIV DNA in HIV-1 infected huNSG mice

Inflammasome activation has been proposed as the main mechanism of CD4+ T cells death during HIV infection (**Doitsh et al., 2010**; **Monroe et al., 2014**). Therefore, we monitored CD4+ T cells percentages and CD4/CD8 T cell ratio longitudinally in the blood and at necropsy in the organs after the administration of the caspase-1 inhibitor. Vehicle and VX-765-treated groups presented a slight but significant reduction of CD4+ T cells percentages and CD4/CD8 T cell ratio in the blood at day 22 post-infection (**Figure 4—figure supplement 2**). Although CD4+ T cells percentages and CD4/CD8 T cell ratio were equivalent in both groups before HIV-1 infection and VX-765 treatment, percentages of circulating CD4+ T cells and CD4/CD8 T cell ratio at day 22 post-infection in the VX-765-treated group were slightly higher (p<0.05; **Figure 5A and B**). At necropsy, we found that CD4+ T cells percentages and CD4/CD8 T cell ratio in the bone marrow but not in the spleen were also slightly higher in the VX-765 treated HIV-1 infected huNSG compared to the vehicle group (p<0.05 and n.s.; **Figure 5C, D, E and F**) indicating a smaller effect of caspase-1 inhibition on CD4 ⁺T cell depletion.

In addition to slightly reducing CD4+ T cell depletion, we observed lower plasma viral loads in VX-765-treated mice (p<0.05; **Figure 6A, B and C**) indicating that inflammasome inhibition interferes with HIV-1 replication. When measuring total HIV-1 DNA in human CD45+ cells from the spleen of all animals, we also found a reduced content of total HIV-1 DNA (**Figure 6D**) in VX-765-treated mice as compared to the vehicle treated mice (1 054 vs 2 889 copies / 10⁶ cells, p=0.029).

## VX-765 inhibits in vivo caspase-1 activity in splenic CD11c⁺ and CD14⁺ cells

To better understand the effect of VX-765 treatment on inflammasome activation in immune cells subsets, we next investigated caspase-1 activity using the FAM-FLICA Caspase-1 assay reagent at day 23 post-infection in CD4+, CD8+, CD14+, and CD11c+ cells from the spleen and the bone marrow in VX-765 and vehicle-treated HIV-1 infected huNSG mice (**Figure 7** and **Figure 7—figure supplement**

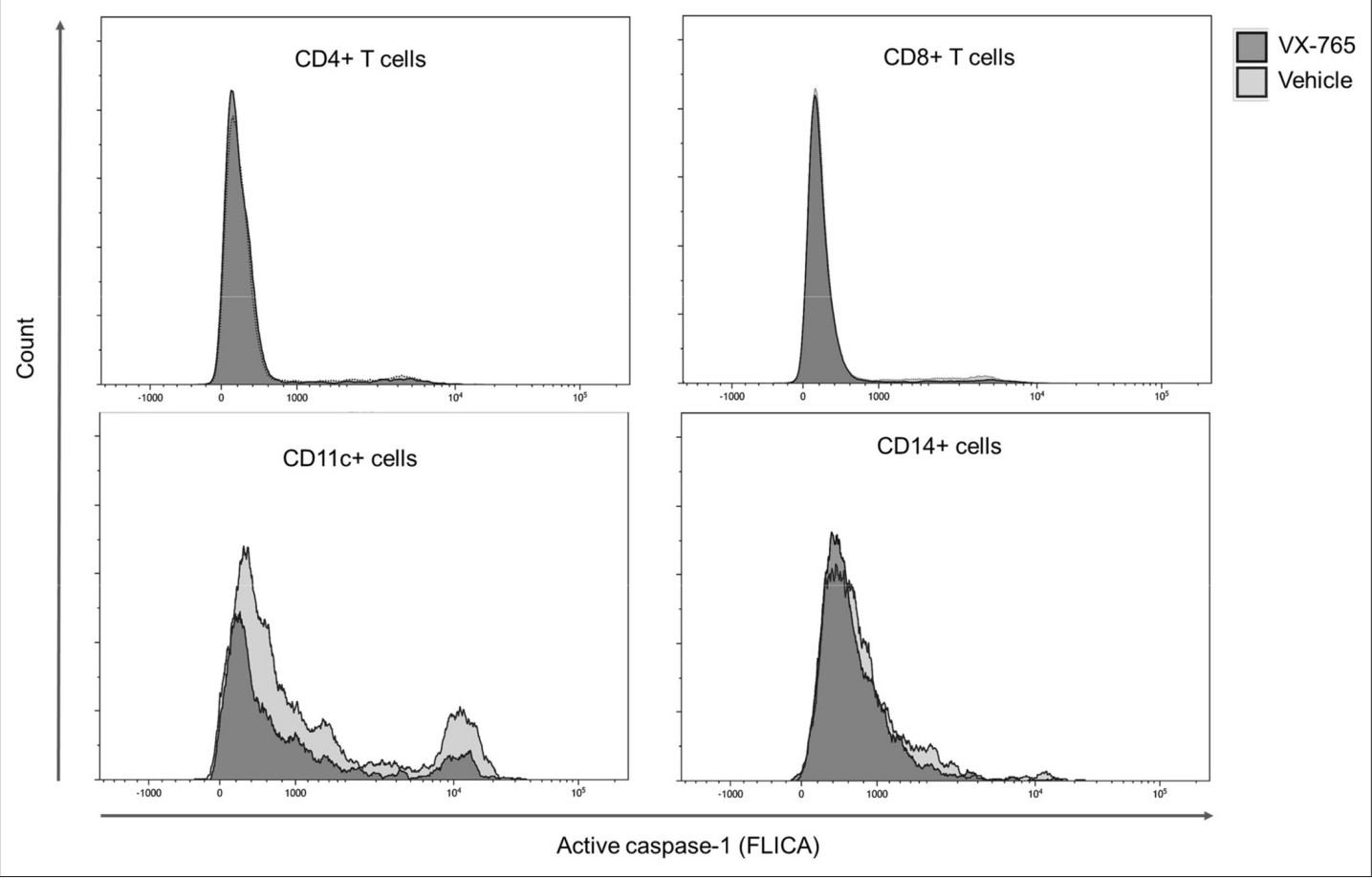

**Figure 7.** Representative histograms of FLICA staining in splenocytes from VX-765 and vehicle-treated HIV-1 infected huNSG mice. Splenocytes were stained for active caspase-1 using FAM-FLICA reagent. Representative MFI of FAM-FLICA+ CD4+, CD8+, CD11c+, and CD14+ splenic cells in one HIV-1 infected huNSG mice treated with vehicle or with VX-765-treated mice at day 22 post-infection is shown.

The online version of this article includes the following figure supplement(s) for figure 7:

**Figure supplement 1.** Gating strategy for FLICA analysis.

*1*). In this assay, the fluorescent FAM-YVAD-FMK probe binds irreversibly to active caspase-1 in living cells. We found the highest caspase-1 activity among CD11c+ and CD14+ cells (*Figure 7*).

Importantly, splenic CD11c+ and CD14+ cells presented a reduced caspase-1 activity upon VX-765 treatment (*Figure 8A, B, C and D*). On the contrary, CD4+ and CD8+ T cells presented low levels of caspase-1 activity and did not respond to VX-765 treatment (*Figure 9A, B, C and D*). In the bone marrow, caspase-1 activity was lower in all subsets compared to the spleen, and none of them presented a decreased caspase-1 activity in response to VX-765 treatment (*Figure 9—figure supplement 1*). These results indicates that inflammasome activation in vivo rather takes places in splenic innate immune CD11c+ and CD14+ cells than splenic adaptive CD4+ or CD8+ or bone marrow immune cells. Furthermore, we found a considerable positive correlation between the percentage of active caspase-1 splenic CD11c+ cells from the vehicle-treated group and the circulating levels of IL-18 (p<0.0001; *Figure 8E*). Such correlations were not significant in other subsets (*Figure 8F* and *Figure 9E and F*) indicating that HIV-1-induced inflammasome activation in splenic CD11c+ cells may be the principal source of circulating IL-18.

In addition to the release of pro-inflammatory cytokines such as IL-1β and IL-18, active caspase-1 initiates the induction of pyroptosis, an highly inflammatory form of cell death (*Monroe et al., 2014*). To determine the effect of inflammasome inhibition on immune cells viability, a fluorescent probe (LIVE/DEAD) was used to identify dying cells in addition to caspase-1 activity. The active caspase-1 (FLICA) fluorescence was plotted against LIVE/DEAD fluorescence to obtain four quadrants: live cells

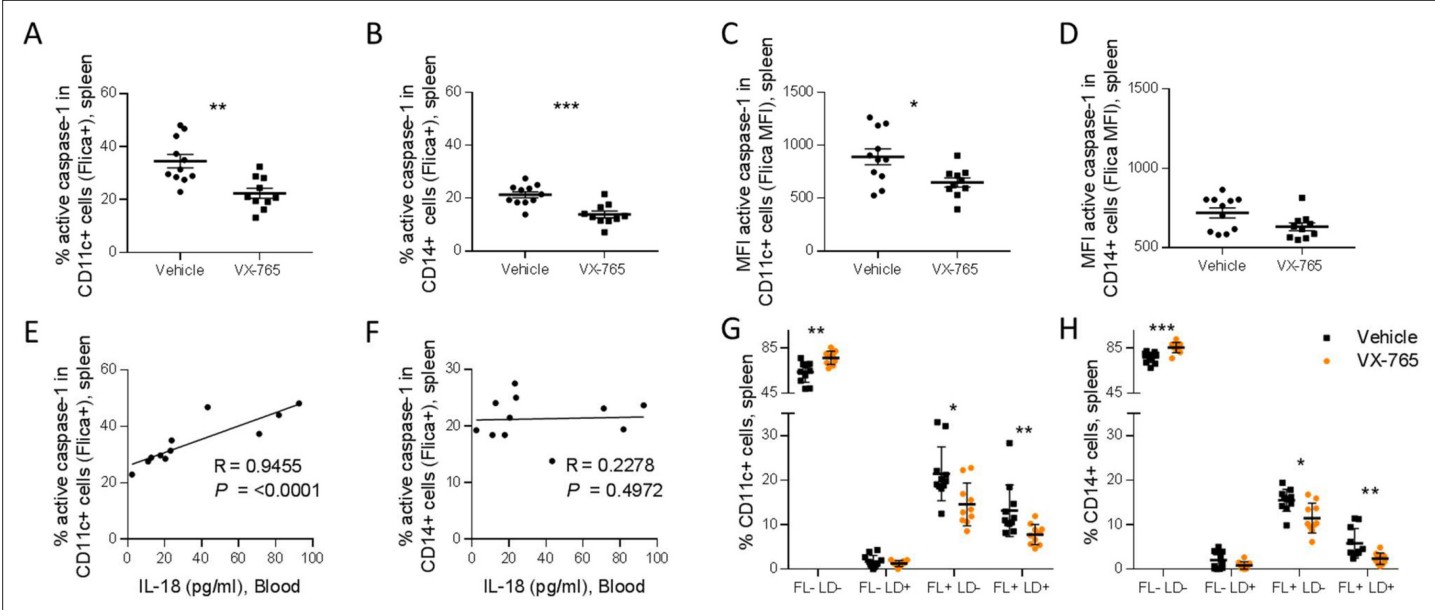

**Figure 8.** VX-765 treatment reduces caspase-1 activation in splenic CD11c⁺ and CD14⁺ in HIV-1 infected huNSG mice. Splenocytes were stained for active caspase-1 using FAM-FLICA reagent. Percentage and MFI of FAM-FLICA⁺ CD11c⁺ (**A, C**), CD14⁺ (**B, D**) splenic cells in vehicle or VX-765 treated HIV-1 infected huNSG mice at day 22 post-infection. Correlation between the percentage of FAM-FLICA⁺ CD11c⁺ (**E**), CD14⁺ (**F**) splenic cells and IL-18 plasma levels in vehicle-treated HIV-1 infect group at day 22 post-infection. Percentage of FAM-FLICA⁻/Live-dead⁻ (FL⁻LD⁻), FAM-FLICA⁻/Live-dead⁺ (FL⁻LD⁺), FAM-FLICA⁺/Live-dead⁻ (FL⁺LD⁻) and FAM-FLICA⁺/Live-dead⁺ (FL⁺LD⁺) CD11c⁺ (**G**), CD14⁺ (**H**) splenic cells in vehicle (Vehicle, n=12) or VX-765 (VX-765, n=12) treated HIV-1 infected huNSG mice at day 22 post-infection. Statistical tests were performed by Mann–Whitney t-tests for comparison of two groups (* p<0.05, ** p<0.005, *** p<0.0005) or Spearman correlation tests.

negative for caspase-1 activity and LIVE/DEAD (FL⁻LD⁻), live cells positive for caspase-1 activity (FL⁺LD⁻), necrotic/late apoptotic cell (FL⁻LD⁺) and pyroptotic cells (FL⁺LD⁺) (**Figure 9—figure supplement 2**). Using theses gates, we observed that VX-765 treatment reduced both the percentages of live cells positive for caspase-1 activity and pyroptotic cells in CD11c⁺ and CD14⁺ when compared to the vehicle-treated group, while the percentage of necrotic/late apoptotic cells remained unchanged (**Figure 8G and H**). In contrast, no changes upon treatment were observed in CD4⁺ and CD8⁺ T cells (**Figure 9G and H**). Given the low levels of caspase-1 activity in CD4⁺ and CD8⁺ T cells together with the observation that caspase-1 activity in these cells was not inhibited using VX-765, a specific caspase-1 inhibitor, we cannot rule out the possibility that FLICA staining in these cells corresponds to a basal background rather than inflammasome activation.

## Discussion

We showed in the current work, that targeting inflammasome activation early after HIV-1 infection using the caspase-1 inhibitor VX-765 might represent a potential therapeutic strategy to improve CD4⁺ T cell homeostasis, and to reduce viral load and immune activation. Many studies have outlined the implications of inflammasome activation during the acute and chronic phase of HIV-1 infection as recently reviewed by *Leal et al., 2020*. The activation of inflammasomes in monocytes, macrophages and dendritic cells leads to the release of cytokines and plays an important part in the first line of host defense against HIV-1 (*Leal et al., 2020*). However, it can also promote viral spread by the recruitment and activation of CD4⁺ T cells (*Klatt and Silvestri, 2012*; *Hunt et al., 2011*). Furthermore, the chronic activation of inflammasomes might be a driver of viral reservoir persistence (*Biancotto et al., 2008*), which in turn can trigger inflammation, creating a pathogenic cycle. Breaking out of this pathogenic cycle could represent an important building block to reduce non-AIDS-related morbidities caused by persisting inflammation during cART (*Deeks et al., 2012*). To date, despite increased therapeutic interests (*Jin et al., 2022*; *Wu et al., 2021*; *Flores et al., 2020*), no compound targeting the inflammasome has been marketed (*Dutartre, 2016*). Nevertheless, the caspase 1 inhibitor VX-765 was approved by the Food and Drug Administration for human clinical trials, showed a good safety

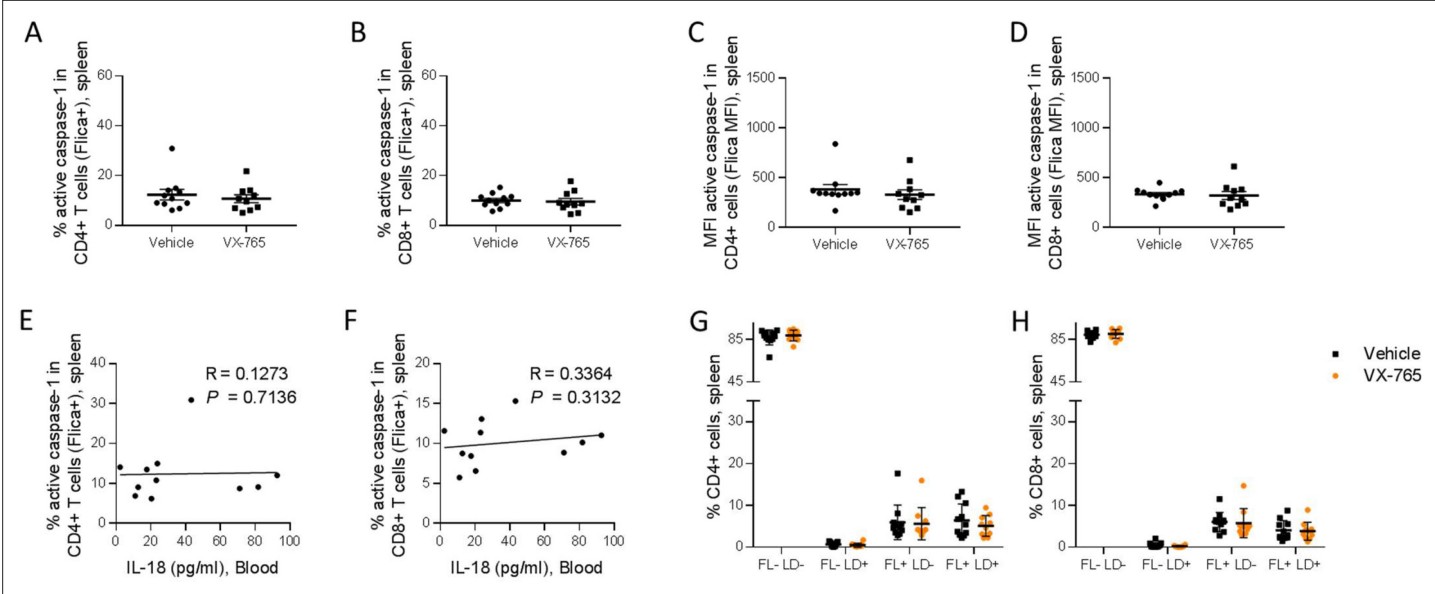

**Figure 9.** VX-765 treatment does not affect caspase-1 activation in splenic CD4+ and CD8+ T cells in HIV-1 infected huNSG mice. Splenocytes were stained for active caspase-1 using FAM-FLICA reagent. Percentage and MFI of FAM-FLICA+ CD4+ (**A, C**), CD8+ (**B, D**) splenic cells in vehicle or VX-765 treated HIV-1 infected huNSG mice at day 22 post-infection. Correlation between the percentage of FAM-FLICA+ CD4+ (**E**), CD8+ (**F**) splenic cells and IL-18 plasma levels in vehicle treated HIV-1 infect group at day 22 post-infection. Percentage of FAM-FLICA-/Live-dead- (FL-LD-), FAM-FLICA-/Live-dead+ (FL-LD+), FAM-FLICA+/Live-dead- (FL+LD-) and FAM-FLICA+/Live-dead+ (FL+LD+) CD4+ (**G**), CD8+ (**H**) splenic cells in vehicle (Vehicle, n=12) or VX-765 (VX-765, n=12) treated HIV-1 infected huNSG mice at day 22 post-infection. Statistical tests were performed by Mann–Whitney t-tests for comparison of two groups or Spearman correlation tests.

The online version of this article includes the following figure supplement(s) for figure 9:

**Figure supplement 1.** VX-765 treatment does not affect caspase-1 activation in the bone marrow.

**Figure supplement 2.** Representative dot plots of FLICA versus LIVE-DEAD staining in splenocytes from VX-765 and vehicle-treated HIV-1 infected huNSG mice.

profile, and represents currently a promising drug to prevent the onset of inflammation in several diseases (**Flores et al., 2020**).

In this regard, we investigated in a first step whether the humanized mouse model might be a suitable pre-clinical model to evaluate anti-inflammasone therapy through the detailed kinetics of inflammasome activation in tissues from humanized mice. During the early host response to HIV-1 infection in humanized NSG mice, we described a clear bi-phasic activation profile of inflammasome-related genes in bone marrow, lungs, lymph node, and spleen. The rapid inflammasome-related transcriptomic changes indicate an early sensing of HIV-1 infection, before HIV-1 RNA was quantifiable in the tissues and may reflect the host response to the virus challenge. These findings are in agreement with previous studies from HIV-1 infection in humans (**Walsh et al., 2014**) and SIV infection in rhesus monkeys (**Barouch et al., 2016**). Interestingly, in the first phase (up to day 3 post-infection), the expression of NLRP3, IL-1β, and IL-18 was significantly upregulated in several organs before viremia was detectable in blood. Of note, in lymph nodes and bone marrow, transcriptomic changes occurred even before detectability of the virus in the respective tissues and despite variable level of human cells engraftment in tissues of humanized mice. The genes induced in the bone marrow included *NLRP3*, able to respond to a diverse set of stimuli including HIV ssRNA and proteins, the adaptor ASC, bridging NLRs such as NLRP3 to the inflammatory Caspase-1(CASP-1) and the downstream pro-inflammatory cytokines IL-1β and IL-18 indicating the up-regulation of the entire NLRP3 pathway. In the lymp nodes, the induction of an early inflammasome from day 1 and day 3 p.i. on, was evident from the upregulation of the NOD-like receptors NLRP1, NLRP3, AIM2, CASP-1, and IL-1β, indicating an early host response to HIV-1 infection in humanized mice, in agreement with results obtained from the SIV model (**Barouch et al., 2016**; **Lu et al., 2016**). After the initial burst we observed a second phase of transcriptomic changes dominated by IFI16 and AIM2 in bone marrow, lymph nodes, and spleen. In response to DNA, IFI16 is able to mediate the induction of IFN-β through the STING

pathway (*Unterholzner et al., 2010*) but it can also form an inflammasome with ASC upon detection of nuclear DNA (*Kerur et al., 2011*). AIM2 is a cytoplasmic double stranded DNA sensor that can initiate the assembly of the inflammasome (*Hornung et al., 2009*; *Bürckstümmer et al., 2009*; *Fernandes-Alnemri et al., 2009*). Previous reports indicated that IFI16 restricts HIV-1 infection trough STING in macrophages (*Jakobsen et al., 2013*) but promotes HIV-1 induced CD4+ T cells death by pyroptosis and thus may drive HIV-1 pathogenesis (*Doitsh et al., 2010*; *Monroe et al., 2014*). Here we show several direct correlations between IFI16 but also AIM2 expression with viral load and the inverse correlations with the percentage of CD4+ T cells (*Figure 2*). These results are reinforced by several correlations obtained between ASC and CASP-1 expression and the percentage of CD4+ T cells in the bone marrow, lymph nodes or in the spleen or with viral load for ASC in the bone marrow 24 days post-infection (*Figure 2—figure supplement 1*). Taken together these data emphasize the dual role of inflammasome at the onset of HIV-1 infection that fights first the virus and then fuels disease progression.

Inflammasome activation by NLRP3 is a tightly regulated process requiring signals in two main steps. First a priming signal which leads to transcription of inflammasome-related genes and secondly, a signal initiating the assembly of the multiprotein complex. As a result, caspase-1 activity promotes the proteolytic cleavage of the pro-inflammatory cytokines IL-1β and IL-18. We report here significant increases of circulating protein levels of IL-18 at day 10 and 24 post-HIV-1 infection (*Figure 3*) indicating a full blown inflammasome activity in humanized mice. We also found a high systemic burst of several plasma cytokines (IFN-γ, TNF-α, IP-10, and IL-18) in agreement with the charateristic cytokine storm of acute HIV-1 infection in humans. For IL-1β, we did not find a detectable and significant increase of the cytokine in the plasma of HIV-1 infected mice as compared to non-infected mice, in contrast to mRNA expression in all tissues. This discrepancy might be due to the short half-life of the cytokine (*Stacey et al., 2009*) and the low basal secretion of protein expression in plasma of the mice. In our humanized mice model, we used an improved engraftment protocol with busulfan myeloablation to reach T cell counts sufficient to sustain long-term HIV replication. Increased CD3+ T cell engraftment levels were obtained, constituted of around 61% of human CD3+ thymocytes, 8% of human monocytes in the bone marrow, and 3% of human myeloid DCs in the spleen, 22 weeks after engraftment of mice (*Singh et al., 2012*). Since IL-1β is mainly produced by macrophages and macrophage-like cells, it is tempting to speculate that less protein is secreted after HIV infection in this model as compared to other animal models. In addition, the cytokine is active at the low ng/mL range or even the pg/mL range in humans or in mice. This low level of secretion could explain its fast consumption or fast degradation in plasma of the humanized mice. In contrast, a clear accumulation of IL-18 was shown, and a clear correlation was however achieved between the level of circulating IL-18 and HIV-1 pathogenesis (viral load and loss of CD4+ T cells, *Figure 3*). In summary, all these data demonstrate that the humanized mouse model of HIV infection recapitulates the mean features of inflammasome activation and is well adapted to assess anti-inflammasome therapies against HIV-1-induced immune activation.

To explore the potential effects of direct inflammasome inhibition in vivo, we next evaluated the caspase-1 inhibitor VX-765 administrated as early as day 2 post-infection and for 21 consecutive days. Daily administration of VX-765 significantly reduced circulating levels of IL-18 and levels of circulating pro-inflammatory cytokines, preserved slightly CD4 +T cell homeostasis, and lowered plasma viral load and HIV-1 reservoirs in human CD45+ immune cells from the spleen. Interestingly, since a low CD4/CD8 T cell ratio is associated with disease progression to AIDS, our results suggest that VX-765 treatment may delay disease progression and preserve CD4+ T cell homeostasis. Pyroptotic CD4+ T cells in lymphoid organs was shown through in vitro models to release pro-inflammatory cytokines, inducing a local inflammation that further recruits new CD4+ T cells to the site of infection. In addition, innate immune cells recruitment and the activation of an NLRP3 inflammasome might enhance the release of pro-inflammatory mediators. This globally enhances HIV-1 replication, thus stimulating the replenishment of the reservoir. It is worthy to note that treatment with VX-765 reduced significantly plasma viral load and the formation of the HIV-1 reservoir in the spleen of humanized mice. This result needs to be further confirmed in central memory and effector memory T cells in a model of HIV-1 latency in humanized mice in combination with cART treatment to explore this potential.

We show here that inflammasome activation in vivo rather takes places in splenic innate immune CD11c+ and CD14+ cells than in the splenic adaptive CD4+ or CD8+ T cells or in the bone marrow

immune cells indicating that HIV-1 induced inflammasome activation in splenic CD11c[+] cells may be the principal source of circulating IL-18. When investigating pyroptosis in subsets of splenic caspase-1 activated cells, we did not find any significant increase in cell death in CD4[+] and CD8[+] T cells. In addition, we did not detect any effect of VX-765 in caspase-1 activation, suggesting that the small caspase-1 activation observed in these subsets might reflect a background activity and could probably be induced earlier by HIV-1 infection, as proposed by Terahara K et al. who showed interestingly that live caspase 1 cells might rapidly die in humanized mice (*Terahara et al., 2021*). Our in vivo data did not confirm previous results obtained in vitro only by the group of *Doitsh et al., 2010* who proposed pyroptosis as the main mechanism leading to CD4[+] T cell depletion after HIV-1 infection. We can not exclude that the assay is not sensitive enough in primary human T cells from humanized mice to show pyroptosis or that we did not catch an earlier best time point. In addition, there is no evidence yet that such microenvironment of abortive infected and bystander CD4[+] T cells can be recapitulated in lymphoid organs of humanized mice where fewer human CD4[+] T cells remains as compared to human lymphoid organs. This could also explain the minimal effect of VX-765 on CD4 [+]T cell depletion as compared to previous results obtained in vitro by *Doitsh et al., 2014* HIV-1 infection is associated with programmed cell death, and NLRP3 inflammasome-mediated immune T cell depletion was nevertheless shown here by different correlations with the dowstream pathways of NLRP3. Through serial necropsies, Barouch et al. demonstrated that, following SIV infection, early host response at the mucosal portal of entry and early sites of distal virus spread involved a robust expression of components of the inflammasome and TGF-β pathways which impairs both innate and adaptative immunities (*Barouch et al., 2016*). Other anti-inflammatory strategies aiming at breaking this vicious circle and reducing HIV persistence are currently investigated in several clinical trials, including the anti-inflammatory, anti-fibrotic angiotensin II blocker losartan (phase 2, NCT01852942) and angiotensin II receptor antagonist telmisartan (phase 1, NCT02170246), immunosuppressors with mTOR inhibitory activities such as sirolimus (phase 2, NCT02440789) and everolimus (phase 4, NCT02429869) or the anti-inflammatory janus kinase inhibitor ruxolitinib (phase 2, NCT02475655). Most of these strategies target inflammation during chronic HIV infection. HIV-1-induced massive systemic immune activation was repeatedly linked to poor outcomes during chronic infection such as the increased risk of cardiovascular complications (*Ross et al., 2009*; *Graham et al., 2013*). Furthermore, levels of cytokines IP-10 and IL-18 remain elevated in cART patients compared to healthy donors and elite controllers (*Côrtes et al., 2018*). Therefore, complementary approaches are needed to reduce persisting immune activation under cART; inhibiting early inflamasomme activation might be a very promising approach that could be combined to cART for treating acute HIV infection. During chronic HIV-1 infection under cART therapy, residual viral replication and viral protein expression from defective provirusses is a continuous driver of inflammation and might also trigger inflammasome activation (*Klatt and Silvestri, 2012*; *Hunt et al., 2011*; *Biancotto et al., 2008*). Furthermore, the incapacity of cART to completely restore immune functions and control chronic inflammation promotes immune aging and the emergence of non-AIDS-related diseases such as cancer, cardiovascular diseases, viral infections, and cognitive decline (*Deeks et al., 2012*). Although cART reduce the aberrant cytokine production of monocytes, some inflammatory monocytes subsist while phagocytic monocytes decrease, resulting in a lower antigen presentation to CD8[+] T cells (*Novelli et al., 2020*). Taken together, all these data suggest that inflamasomme inhibition, by offering simultaneous reduction of inflammation and affecting HIV-1 replication, might be further evaluated to support cure research.

## Ideas and speculations

Besides HIV-1, a number of viruses such as influenza or measles have evolved mechanisms to suppress the NLRP3 inflammasome (*Choudhury et al., 2021*). More recently, it has been proposed that inhibiting the inflammasomes pathways and/or pyroptosis in COVID-19 therapy would be an efficient strategy to decrease SARS-CoV-2 virulence and NLRP3-mediated inflammation for treatment of severe COVID-19 disease (*Freeman and Swartz, 2020*). NLRP3 modulation might therefore be a promising intervention to balance a functional immune response for infection control of different viruses. Novel mechanisms for NLPR3 modulation have been aslso proposed to promote an effective innate immune response against fungal pathogens (*Briard et al., 2020*), highlighting that further investigations need to be conducted to understand the adapted control of NLRP3. Ultimately, our results demonstrate the benefits of inhibiting inflammasome early after HIV-1 infection and suggest to evaluate such therapies

in combination with early cART. Since administration of cART will be less intrusive with long-acting formulations and combined with other therapies in the context of an HIV cure in the near future, we and others (*Ekabe et al., 2021*) strongly support the evaluation of inflammasome inhibitors for HIV-infected patients.

# Materials and methods

## Humanized mice and HIV infection

NOD.Cg-*Prkdc^scid^ Il2rg^tm1Wjl^*/SzJ (NSG) (005557) mice were purchased from the Charles River Laboratory (France). All mice were bred and maintained in microisolator cages at the specific pathogen free (SPF) animal facility of the Luxembourg Institute of Health according to national and EU regulations. All experiments on animals were performed with the authorizations from the animal welfare committee of the Luxembourg Institute of Health and the Ministry of Veterinary and Agriculture of Luxembourg (protocol DII-2017–12). 3- to 4-weeks-old mice juvenile NSG mice were conditioned and humanized as previously described (*Adams et al., 2021*). Briefly, NSG mice were conditionned with two intraperitoneal injections of Busulfan (2x20 mg/kg, Busilvex) with a 12 hr interval. Conditionned NSG mice were humanized with CD34⁺ hematopoietic stem cells isolated from human cord blood (CB) using a magnetic activated cell sorting CD34⁺ progenitor cell isolation kit (Stem Cell Technologies, Belgium). CB was provided by the Cord Blood Bank Central Hospital University (Liège, Belgium) and was collected after obtaining written infomed consent. The protocol (reference 1513) was accepted by the ethics committee of the University hospital of Liège (reference B70720072580). 2x10⁵ freshly isolated CD34⁺ cells were transplanted intravenously (i.v.). Twenty four weeks post-transplantation, humanized mice with more than 10% human CD45⁺ cells in the peripheral blood were transferred to level 3 animal facilities and infected by intraperitoneal injection of the HIV-1 laboratory adapted strain JRCSF (ARP-2708 HIV-1Strain JR-CSF Infectious Molecular Clone pYK-JRCSF, NIH HIV reagent program, 10.000 TCID50). An appropriate sample size (n=5) was calculated during the study design to obtain groups with a difference of humanisation of 10% by taking into account a common standard deviation of 5% using a bilateral student t-test based on a 95% confidence level and homogeneous viral load (*Adams et al., 2021*). The level of humanization and viremia was randomized between the groups.

## VX-765 treatment

VX-765 (Invivogen) was dissolved in 20% cremophor and injected intraperitoneally (i.p.) in HIV-1 infected humanized mice at 200 mg/kg once a day starting at day 2 post-infection and for 21 consecutive days. Control humanized mice received the corresponding vehicle. An appropriate sample size (n=12) was calculated during the study design for each group to obtain a difference on viral load of 15% between the groups by taking into account a common standard deviation of 10% using a bilateral student t-test based on a 95% confidence level. The level of humanization was randomized between the two groups before HIV-1 infection.

## Sampling collection and cell isolation

Whole blood was collected into EDTA-coated tubes (BD Microtainer K2E tubes; BD Biosciences) from the temporal vein for longitudinal sampling or by cardiac puncture at necropsy. Plasma was separated from cells by centrifugation 10 min at 2000 rpm at 4°, aliquoted in sterile Eppendorf tubes and stored at –80 °C until subsequent analysis. Peripheral blood cells were further stained for flow cytometry. Organs were collected at necropsy, washed in cold RPMI and processed immediately for cell isolation. Single-cell suspensions were prepared from spleens, lymph nodes, and bone marrows. Spleens and lymph nodes were gently disrupted in cold PBS using the plunger end of a syringe. Bone marrow cells were obtained from tibias and femurs. Both ends of the bones were cut and the bone marrow were flushed using a 25-gauge needle and 1 mL syringe filled with cold PBS. Cells from bone marrow and splenocytes suspensions were passed through 70 µm nylon cell strainers (BD biosciences) and centrifuged 10 min at 1200 rpm at 4 °C. Single-cell suspensions were further used for hCD45⁺ cells isolation, stained for flow cytometry or pelleted prior to DNA extraction. Each sampling was further blind-tested and blind-analysed.

## Human CD45⁺ cells isolation, RNA extraction and reverse transcription polymerase chain reaction

Human CD45$^+$ cells were isolated from peripheral blood or spleen, bone marrow and lymph nodes single-cell suspensions using CD45 MicroBeads, human positive selection kit (Miltenyi Biotec) and total RNA was extracted from hCD45$^+$ cells using NucleoSpin RNA kit (Macherey-Nagel) following manufacturer's instructions using an input of 100 µl of cell lysates. HIV-1 RNA was measured by digital droplet PCR (ddPCR) assay as previously described (*Adams et al., 2021*). A total of 200 ng of RNA was reverse transcribed into cDNA using High-Capacity RNA-to-cDNA Kit (Applied Biosystems). The cDNA was used for the quantification of human *NLRP1* (TaqMan Assay: Hs00248187_m1), *NLRP3* (TaqMan Assay: Hs00918082_m1), *NLRC4* (TaqMan Assay: Hs00892666_m1), *AIM2* (TaqMan Assay: Hs00915710_m1), *IFI16* (TaqMan Assay: Hs00986757_m1), *ASC* (TaqMan Assay: Hs00203118_m1), *CASP1* (TaqMan Assay: Hs00354836_m1), *IL1B* (TaqMan Assay: Hs01555410_m1), and *IL18* (TaqMan Assay: Hs01038788_m1) genes expression by real-time PCR. *ACTB* gene (TaqMan Assay: Actb-Hs00357333_g1) was used as a control for expression levels normalization. PCR amplification of cDNA was done in 20 µL volume containing 10 µL of TaqMan Fast Advanced Master Mix (2 X) (Applied Biosystems), 1 µL of TaqMan Assay (20 X) (Applied Biosystems), 7 µL of Nuclease-Free Water and 2 µL of cDNA. All quantitative PCR were performed on a 7500 Real-Time PCR System (Applied Biosystem). The results were analyzed using the $2^{-\Delta\Delta CT}$ method.

## Flow cytometry

Human hematopoietic cell engraftment and immunophenotyping in mouse xenorecipients was performed by flow cytometry on peripheral blood or spleen and bone marrow single-cell suspensions at various timepoints. Red blood cells from peripheral blood and spleen samples were lysed using RBC lysis buffer (BD). Cells were stained for 30 min at 4 °C in 5 ml round bottom polypropylene tubes with fluorochrome-labeled monoclonal antibodies. The humanisation antibody cocktail contained CD4-BUV395 (SK3), CD3-BUV496 (UCHT1), CD8-BUV805 (SK1), hCD45-BV421 (HI30), mCD45-PE Cy5 (30-F11), CD14-APC (M5E2) from BD Bioscience and CD19-PE (SJ25C1) from Biolegend. For phenotyping studies CD152-BV786 (BNI3), CD69-PE CF594 (FN50) from BD and HLA DR-BV711 (L243), CD38-PerCP Cy5.5 (HIT2), CD279-PE Cy7 (EH12.2H7) from Biolegend were added to the cocktail. For the FAM-FLICA Caspase-1 assay (ImmunoChemistry Technologies), 30 X FAM-FLICA fluorescent dye was added on cells at a final ratio of 1:60 and incubated 1 hr at 37 °C and washed with 2 ml of 1 X Apoptosis wash buffer. For this experiment, CD11c-PE (B-ly6) (BD), CD163-PerCP Cy5.5 (GHI/61), and CD16-PE Cy7 (3G8) (Biolegend) were added for flow cytometry staining. For all experiments, cell viability was assessed using LIVE/DEAD Near-IR dead cell stain kit from Invitrogen according to the manufacturer's instructions. Cells were fixed for 1 hr at 4 °C using 1 X BD Lysing solution (Cat n°: 349202) before acquisition. Samples were acquired on a FACS Fortessa SORP 5 laser instrument (BD Biosciences) and analyzed with the Kaluza Flow Cytometry Analysis Software (Beckman Coulter).

## Total HIV-1 DNA quantification and plasma viral load

Total DNA was extracted from spleen, lymph nodes and bone marrow cells by manual extraction with the NucleoSpin Tissue kit (Macherey-Nagel) according to manufacturer's instructions. Aliquots of eluted samples were frozen at −80 °C until total HIV-1 DNA quantification by ddPCR as previously described (*Adams et al., 2021*). Plasma viral load was measured by digital droplet PCR (ddPCR) assay as previously described (*Adams et al., 2021*). Thirty µL of humanized mice plasma was diluted with 70 µL of PBS and processed. The limit of detection in this assay is 235 copies/ml.

## Cytokine levels

Levels of human cytokines IL-1β, IL-6, IL-18, IFN-γ, IP-10, and TNF-α were quantified in 25 µL of plasma using a multiplex U-PLEX Biomarker assay kit (Meso Scale Diagnostics) according to the manufacturer's instructions. The Lower Limit Of Detections (LOD) were 0.15 pg/mL for IL-1β, 0.33 pg/mL for IL-6, 0.5 pg/mL for IL-18, 1.7 pg/mL for IFN-γ, 0.49 pg/mL for IP-10, and 0.51 pg/mL for TNF-α. Data were acquired and analysed using MESO QuickPlex SQ 120 instrument.

## Statistical analysis

Statistical analysis was performed using GRAPHPAD PRISM software. Data were expressed as the mean value ± SEM. Groups were compared using unpaired Mann-Whitney $U$ test or Wilcoxon's matched pairs signed rank test. A p-value <0.05 was considered to be significant. Two-tailed Spearman correlation coefficient (r) was calculated, with resulting p-values at 95% confidence.

## Acknowledgements

We acknowledge Charlène Vershueren and Quentin Etienne for their technical assistance in animal experimentation. The following reagent was obtained through the NIH HIV Reagent Program, Division of AIDS, NIAID, NIH: Human Immunodeficiency Virus 1 (HIV-1), Strain JR-CSF Infectious Molecular Clone (pYK-JRCSF), ARP-2708, contributed by Dr Irvin S Y Chen and Dr Yoshio Koyanagi."

## Additional information

### Funding

| Funder | Grant reference number | Author |
| --- | --- | --- |
| Fonds National de la Recherche Luxembourg | AFR PhD ID 10111126 | Philipp Adams |
| Fonds National de la Recherche Luxembourg | Next Immune DTU PRIDE ID 11012546 | Rafaela Schober |

The funders had no role in study design, data collection and interpretation, or the decision to submit the work for publication.

### Author contributions

Mathieu Amand, Conceptualization, Resources, Data curation, Formal analysis, Supervision, Investigation, Methodology, Writing – original draft, Project administration; Philipp Adams, Conceptualization, Resources, Formal analysis, Investigation, Writing – original draft; Rafaela Schober, Resources, Data curation, Formal analysis, Investigation, Writing - review and editing; Gilles Iserentant, Resources, Formal analysis, Visualization, Methodology, Writing - review and editing; Jean-Yves Servais, Resources, Formal analysis, Writing - review and editing; Michel Moutschen, Conceptualization, Investigation, Methodology, Writing - review and editing; Carole Seguin-Devaux, Conceptualization, Formal analysis, Supervision, Funding acquisition, Validation, Investigation, Methodology, Writing – original draft, Project administration

### Author ORCIDs

Carole Seguin-Devaux (iD) http://orcid.org/0000-0003-0636-5222

### Ethics

The study was peformed at the Luxembourg Institute of Health in accordance with national and EU regulations with approved institutional animal care and use committee protocols . All experiments on animals were performed with the authorizations from the animal welfare committee of the Luxembourg Institute of Health and the Ministry of Veterinary and Agriculture of Luxembourg (protocol DII-2017-12).

### Decision letter and Author response

Decision letter https://doi.org/10.7554/eLife.83207.sa1
Author response https://doi.org/10.7554/eLife.83207.sa2

## Additional files

### Supplementary files
• MDAR checklist

## Data availability

The data have been uploaded in the Zenodo repository.

The following dataset was generated:

| Author(s) | Year | Dataset title | Dataset URL | Database and Identifier |
|---|---|---|---|---|
| Mathieu A, Philipp A, Rafaëla S, Gilles S, Yves SJ, Michel M, Carole SD | 2023 | The anti-caspase 1 inhibitor VX-765 reduces immune activation, CD4+ T cell depletion, viral load and total HIV-1 DNA in HIV-1 infected humanized mice | https://zenodo.org/record/7600354#.Y-5QsuzP23I | Zenodo, 7600354#.Y-5QsuzP23I |

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
