## [Editor Report]

This important study examined the induction of inflammasome activation by HIV infection in a humanized NSG mouse model. The authors convincingly show that inflammasome activation plays a key role in CD4 T cell depletion and can be inhibited by the anti-caspase 1 inhibitor VX-765. The results are of interest to scientists and physicians interested in the treatment and pathogenesis of HIV-1 infection.

---

## [Decision Letter]

**Decision letter after peer review:**

Thank you for submitting your article "The anti-caspase 1 inhibitor VX-765 reduces immune activation, CD4 + T cell depletion, viral load and total HIV-1 DNA in HIV-1 infected humanized mice" for consideration by *eLife*. Your article has been reviewed by 2 peer reviewers, and the evaluation has been overseen by a Reviewing Editor and Satyajit Rath as the Senior Editor. The reviewers have opted to remain anonymous.

The study examined the induction of inflammasome activation by HIV infection in a humanized NSG mouse model. The results suggest that inflammasome activation plays a key role in CD4 T cell depletion and can be inhibited by the anti-caspase 1 inhibitor VX-765. The results are suggestive but not all findings seem entirely consistent and some issues regarding the correlation analyses, significance of some cytokine measurements, and inability to detect caspase-1 activity in CD4 and CD8 T cells need to be addressed.

Essential revisions:

1. How do the authors explain lack of plasma viremia despite high levels of HIV RNA in all tissues and blood at day 10 in two of the animals. Please also address the inconsistencies in inflammasome gene and IL-18 upregulation across tissues and timepoints noted by reviewer 1.

2. Are the correlations shown in Figure 2 (and figure S4) still significant if only infected animals are included?

3. The levels of some cytokines shown in figure 3 and figure 4 seem to be below the usual limit of detection for the assay used. This should be clarified and only reliable measurements included in correlation analyses.

4. It is surprising that no caspase-1 activity could be detected in CD4 and CD8 T cells. The authors should try to use a more sensitive assay, if available. If caspase-1 activity is not increased in CD4 T cells after HIV infection in this mouse model differences to previous data using human tissues and the overall hypothesis that inflammasome activity contributes to viral persistence in CD4 T cells need more discussion.

5. The protective effect on CD4 T cells is modest and should be presented with caution. In addition, the discussion about HIV reservoirs should be shortened or (better) the effect of VX-765 on the HIV reservoirs e.g. in central and effector memory CD4 T cells should be determined to strengthen this part.

6. It should be examined whether administration of the anti-caspase 1 inhibitor VX-765 suppresses the induction of inflammasome-associated genes upon HIV infection in this mouse model.

*Reviewer #1 (Recommendations for the authors):*

Specific comments:

1. The manuscript contains numerous typos and grammatical errors and should be extensively spellchecked before resubmission.

2. It is surprising to see high levels of HIV RNA in all tissues and blood at day 10 but no detectable plasma viremia in 40% (2 out of 5) of the animals. The authors should comment on this discrepancy.

3. There appear to be inconsistencies in inflammasome gene upregulation across tissues and timepoints. Is there a biological explanation for this or is it due to inconsistencies in the qPCR assay? Furthermore, IL-18 gene transcription only appears to be upregulated in the BM at D3 post infection and in none of the other tissues, particularly at D24 post-infection. This seems quite different from the cytokine levels measured in Figure 3, where IL-18 is 10-fold higher at D24 than at pre-infection. What would be the reason for this discrepancy?

4. It is shown clearly that IFI16 and AIM2 expression are increased after infection, and this fits with the hypothesis and previous literature showing inflammasome activation after HIV infection. However, the correlations in Figure 2 (and figure S4) appear quite misleading, as they seem to be solely driven by the inclusion of data from uninfected animals. These correlations need to be shown only including infected animals, as there is a large enough heterogeneity in the viral loads and CD4 counts of these animals to appreciate any true associations.

5. There appear to be issues with the measurement of a number of the cytokines shown in figure 3 and figure 4. While the authors do not describe the limit of detection for the assay used, they indicated they are using Mesoscale U-PLEX biomarker assay kit. The listed LOD for this kit includes IL-1B at 0.15 pg/mL, IL-6 at 0.33 pg/mL and TNF-a at 0.51 pg/mL. It appears that the large majority, if not all, of the measurements of these 3 cytokines are below the limit of detection. To display these data points as accurate values without any indication of this is incorrect. If this is the true LOD, all data points below that limit should be clearly indicated and should not be used in correlation analyses without imputed values. If the authors have an alternative LOD than that listed for the commercially available kit, it needs to be made clear what that limit is and how it was calculated. It is concerning that the primary cytokine involved in caspase-1 activation of the inflammasome (as described in line 103) is unable to be measured in this manuscript.

6. Figure 5 could be improved by clearly indicating the day post-infection the data for panels B, C, E and F represent. Furthermore, panel E in figure 5 is indicated to show Blood in the y axis label, but BM in the figure legend.

7. The authors should clarify the issue with the lack of caspase-1 activity in CD4 and CD8 T cells detected. If this is a problem with the assay, a different approach needs to be used. If caspase-1 activity is not increased in CD4 T cells after infection in their model, this data impacts the overall hypothesis that inflammasome activity contributes to viral persistence in CD4 T cells and requires a larger discussion.

8. The impact on CD4 T cell preservation seems minimal. While there is a statistically significant difference in the blood, it appears to be very slight. Furthermore, this difference does not extend to the spleen. This should be acknowledged.

9. The Y axes in figure 6 panels A, B, and C should all be shown in log scale, with no split axes and should all be equivalent to each other as is standard for displaying plasma viral loads in HIV studies.

10. There is much discussion of the importance of limiting the HIV reservoir and doing so in combination with early-initiated antiretroviral therapy. This discussion should be reduced, as this manuscript does not follow animals beyond day 22 post-infection and does not include animals receiving ART.

*Reviewer #2 (Recommendations for the authors):*

1. The Authors explained well in the introduction the elimination of HIV reservoirs as a key factor for HIV cure. Also, the Authors reported anti-caspase 1 inhibitor VX-765 reduces HIV reservoirs. However, in the study the Authors did not quantify the different HIV reservoirs (For example Central memory and effector memory CD4 T cells) and the effect of VX-765 on the population of these HIV reservoirs. It will be helpful if the authors could illustrate if VX-765 reduces the population of HIV reservoirs

2. The authors nicely used a negative control for the anti-caspase 1 inhibitor VX-765 during the study. However, there was no positive control. The design of the experiment can be improved by including a positive control like anti-inflammatory drug.

3. The authors nicely illustrated the expression of genes associated with inflammasome activation in HIV infection. However, this was not done after administration of the anti-caspase 1 inhibitor VX-765. It will make more sense to also illustrate the changes in the transcriptomic profile of markers of inflammasome activation after treatment with VX-765.

4. Studies have revealed that bystander CD4 T pyroptosis account for about 95% depletion of CD4 T cells in HIV infection https://doi.org/10.1016/j.cell.2010.11.001. However, in this study it was shown that there was more inflammasome activation in infected than uninfected CD4 T cells (Figure 2). Could the Authors expand more of this?

---

## [Author Response]

Reviewer #1 (Recommendations for the authors):Specific comments:1. The manuscript contains numerous typos and grammatical errors and should be extensively spellchecked before resubmission.

The manuscript has been reviewed and typos and grammatical errors have been corrected.

2. It is surprising to see high levels of HIV RNA in all tissues and blood at day 10 but no detectable plasma viremia in 40% (2 out of 5) of the animals. The authors should comment on this discrepancy.

As explained in the section Material and Methods, HIV-1 mRNA was measured in mice using plasma (Figure 1—figure supplement 1) and using RNA extracted from CD45 + cells isolated from tissues or blood (Figure 1—figure supplement 2). For plasma viral load, a maximum of 30 µl whole blood was collected from the temporal vein for longitudinal sampling. 30 µl of mice plasma were diluted in 70 µl of PBS for RNA extraction and RNA extracted as described in P Adams et al. (iScience Volume 24, Issue 1, 22 January 2021). Human CD45 + cells were isolated from peripheral blood (collected by cardiac puncture at necropsy) or spleen, bone marrow and lymph nodes single cell suspensions using CD45 MicroBeads, and total RNA was extracted using an input of 100 µl of cell lysates after purification. Therefore, the input for RNA extration is much lower in plasma than in cell lysates or in blood and this explain the discrepancy between the two measurements. This is now detailed in the revised material and methods. In addition, we used a protocol of humanisation with busulfan myeloablation regimen and transplantation of CD34 + cell isolated from cord blood in 3–4 week old NSG mice. As shown in Author response image 1, the level of engraftment was more important in tissues than in peripheral blood (human CD45 ^+^ cells measured by flow cytometry when organs were collected). This explain why HIV-mRNA was detected earlier in human CD45 + isolated from tissues than from blood as soon as day 3. The 2 animals that displayed no detectable viremia at day 10 showed also less HIV-1 mRNA in human CD45 ^+^ cells isolated from blood and tissues (NS127 and NS136). Our results are in agreement with Barouch et al. (Cell 2016) which reported a similar earlier HIV-1 mRNA detectability in tissues as compared to blood in a SIV infection model using Rhesus monkeys.

**Author response image 1. sa2fig1:** Mean of percentage (± SD) of human CD45^+^ cells in blood, Lymph node (LN), Bone Marrow (BM), lung and spleen of humanized mice before (day 0) and after HIV infection (day 3, 10 and 24).

3. There appear to be inconsistencies in inflammasome gene upregulation across tissues and timepoints. Is there a biological explanation for this or is it due to inconsistencies in the qPCR assay?

Inflammasome upregulation was measured at day 1, 3, 10 and 24 after HIV infection from human CD45^+^ cells isolated from the different tissues coming from different mice pooled for each day of measure (D0, D1, D3, D10, D20) and sacrificed. Therefore, the results obtained at day 1, 3 , 10 or 24 are not longitudinal samples coming from the same mice but from different mice and could explain the variabiliy and inconsistencies in inflammasome gene upregulation. Indeed, HIV-1 replication is quite variable among the mice and induced a loss of CD4^+^ T cells that is also variable among the tissues and mice in time (as shown in Author response image 2). This is also the case for the number of CD8^+^ T cells in tissues from the different mice (Author response image 3). This number is variable and based on the level of engraftment of each mice in each tissue, similarly to the other human immune cells producing the different inflamasomme genes or cytokines. All these reasons can explain the inconsistencies observed in Figure 1—figure supplement 3 of the manuscript that are not due in the qPCR assay.

**Author response image 2. sa2fig2:** Mean percentage (± SD) of human CD4^+^ T cells in human CD45^+^CD3^+^ cells in blood and tissues of humanized mice before (D0) and after HIV infection (D1, D3, D10 and D24).

**Author response image 3. sa2fig3:** Mean percentage (± SD) of human CD8^+^ T cells in human CD45^+^CD3^+^ cells in blood and tissues of humanized mice before (D0) and after HIV infection (D1, D3, D10 and D24).

Furthermore, IL-18 gene transcription only appears to be upregulated in the BM at D3 post infection and in none of the other tissues, particularly at D24 post-infection. This seems quite different from the cytokine levels measured in Figure 3, where IL-18 is 10-fold higher at D24 than at pre-infection. What would be the reason for this discrepancy?

Il-18 has unique characteristics that could explained why we have observed only an early upregulation of its mRNA at day 3 in the bone marrow and then a net accumulation in plasma untill day 24. The cytokine is synthesized as an inactive precursor requiring processing by caspase-1 into an active cytokine. Importantly, in contrast to IL-1β, the IL-18 precursor is constitutively present in nearly all cells in healthy humans and animals (Novick D et al., Interleukin-18, more than a Th1 cytokine, Seminars in Immunology, 2013, Volume 25, Issue 6). Many cell types produce IL-18 including hematopoietic cells and circulating monocytes, resident macrophages, and DCs. The IL‐18 precursor is also found to be released by endothelial cells, keratinocytes, osteoblasts, most intestinal epithelial cells throughout the entire gastrointestinal tract, and mesenchymal cells. IL‐18 can also be discharged in its precursor form from dead cells which can be acted upon by neutrophil proteases such as proteinase most 3 into its active form (Interleukin-18 cytokine in immunity, inflammation, and autoimmunity: Biological role in induction, regulation, and treatment, Amarachi Ihim S. et al. Front. Immunol., 11 August 2022). Peritoneal macrophages and mouse spleen also contain the IL-18 precursor. In the absence of disease, the IL-18 precursor is constitutively expressed in endothelial cells, keratinocytes and intestinal epithelial cells. Macrophages and dendritic cells are the primary sources for the release of active IL-18, and this could explain why we found only an early mRNA upregulation of Il-18 in the bone marrow coming from hematopoietic cells and then the IL-18 cytokine is released in plasma upon activation by the cells containing the constitutive precursor. Indeed, IL-18 requires 10–20 ng/mL and sometime higher levels to activate target cells as compared to IL-1β that is active on cells in the low ng/mL range and often in the pg/mL range (Interleukin-18 cytokine in immunity, inflammation, and autoimmunity: Biological role in induction, regulation, and treatment, Amarachi Ihim S. et al. Front. Immunol., 11 August 2022). Taken together, these data support the strike difference between IL-18 and Il-1β concentrations measured in humanized mice after HIV infection and the fact that Il-18 accumulated in plasma and not at the gene level in the spleen or the lymp nodes. A sentence were added in the results part of the manuscript and in the discussion to explain this.

4. It is shown clearly that IFI16 and AIM2 expression are increased after infection, and this fits with the hypothesis and previous literature showing inflammasome activation after HIV infection. However, the correlations in Figure 2 (and figure S4) appear quite misleading, as they seem to be solely driven by the inclusion of data from uninfected animals. These correlations need to be shown only including infected animals, as there is a large enough heterogeneity in the viral loads and CD4 counts of these animals to appreciate any true associations.

The correlations have been now performed by including the mice showing evidence of infection with detectable HIV-RNA in tissues or in blood, and several correlations remained significantly significant. Figure 2 and Figure 2—figure supplement 1 have been updated accordingly with the new graphs of correlations.

5. There appear to be issues with the measurement of a number of the cytokines shown in figure 3 and figure 4. While the authors do not describe the limit of detection for the assay used, they indicated they are using Mesoscale U-PLEX biomarker assay kit. The listed LOD for this kit includes IL-1B at 0.15 pg/mL, IL-6 at 0.33 pg/mL and TNF-a at 0.51 pg/mL. It appears that the large majority, if not all, of the measurements of these 3 cytokines are below the limit of detection. To display these data points as accurate values without any indication of this is incorrect. If this is the true LOD, all data points below that limit should be clearly indicated and should not be used in correlation analyses without imputed values. If the authors have an alternative LOD than that listed for the commercially available kit, it needs to be made clear what that limit is and how it was calculated.

The LOD of the Mesoscale assay (Author response table 1, LOD of the manufactor) have been now added in the material and methods and all the measures being under the LOD have been excluded in the analyses. Therefore, only the graphs and correlations with a sufficient number of detectable measures for statistical analyses are now shown in Figure 3, Figure 3—figure supplement 1, Figure 4, and Figure 4—figure supplement 1.

**Author response table 1. sa2table1:** LOD of the Mesoscale U-PLEX biomarker assay kit for the different cytokines.

U-PLEX Assay	LLOD(Sensitivity)	Dynamic Range
Human IFN-γ	1.7 pg/mL	1.7-17,000 pg/mL
Human IL-1β	0.15 pg/mL	0.15-3,800 pg/mL
Human IL-2	0.70 pg/mL	0.70-1,900 pg/mL
Human IL-6	0.33 pg/mL	0.33-2,000 pg/mL
Human IL-12p70	0.69 pg/mL	0.69-5,3000 pg/mL
Human IL-18	0.5 pg/mL	0.5-14,000 pg/mL
Human IP-10	0.49 pg/mL	0.49-6,000 pg/mL
Human MIP-1β	1.5 pg/mL	1.5-1,600 pg/mL
Human TNF-α	0.51 pg/mL	0.51-3,700 pg/mL
Human TRIAL	0.66 pg/mL	0.66-10,000 pg/mL

It is concerning that the primary cytokine involved in caspase-1 activation of the inflammasome (as described in line 103) is unable to be measured in this manuscript.

As mentioned above, IL-1β activate target cells in the low ng/mL range and often in the pg/mL range (Interleukin-18 cytokine in immunity, inflammation, and autoimmunity: Biological role in induction, regulation, and treatment, Amarachi Ihim S. et al. Front. Immunol., 11 August 2022) in human or animals. In addition, in healthy human subjects and also in healthy mice, gene expression for IL-1β in blood mononuclear cells and hematopoietic cells is absent and there is no evidence that the IL-1β precursor is constitutively present in epithelial cells (Interleukin-18 cytokine in immunity, inflammation, and autoimmunity: Biological role in induction, regulation, and treatment, Amarachi Ihim S. et al. Front. Immunol., 11 August 2022). To our knowledge, the MSD assay was the most sensitive and quantitative available test (dynamic range between 0.15 and 3.8 pg/ml), but was not sensitive enough to catch a detectable and significant increase of Il-1β in humanized mice infected with HIV. This could be a combined consequence of a very low level of secretion of IL-1β by the engrafted human cells and the short half life of IL-1β in plasma of the mice. The cytokine being active at the low ng/mL range or even the pg/mL range, this could explain its fast consumption or degradation in plasma. We also tried western-blotting for Il-1 β, but we could not retrieve enough of proteins from human CD45+ cells isolated from the tissues of the humanized mice to evaluate its expression.

We added two sentences in the result part and in the discussion part to clarify the absence of IL-1β in plasma of the mice.

6. Figure 5 could be improved by clearly indicating the day post-infection the data for panels B, C, E and F represent. Furthermore, panel E in figure 5 is indicated to show Blood in the y axis label, but BM in the figure legend.

The figure 5 has been changed accordingly and mention now Day 22 post-infection above each graph. Panel E is indeed the panel of bone marrow, the name Blood was an error and was changed to BM in the revised Figure 5.

7. The authors should clarify the issue with the lack of caspase-1 activity in CD4 and CD8 T cells detected. If this is a problem with the assay, a different approach needs to be used. If caspase-1 activity is not increased in CD4 T cells after infection in their model, this data impacts the overall hypothesis that inflammasome activity contributes to viral persistence in CD4 T cells and requires a larger discussion.

To measure the Caspase-1 activity in the bone marrow and lymph nodes of humanized mice, we used the fluorescent labeled inhibitors of caspase-1 (FLICA) technique to stain active caspase -1 with a FAM probe. We used the same probe as in the work of Doitsh G. et al. in 2014 who demonstrated the role of caspase-1 in ex vivo human lymphoid aggregate culture (HLAC) system formed with fresh human tonsil or spleen tissues, to reflect HIV-1 infection by a small number of viral particles without mitogens in a lymphoid microenvironment. Infection of these cultures with HIV-1 produces extensive loss of CD4 T cells, but, as described by the authors, >95% of the dying cells are abortively infected with HIV reflecting their nonpermissive, quiescent state. Flow cytometry is the most sensitive tool that we could use in our humanized model of HIV-1 infection to detect caspase-1 activity. The Caspase-Glo 1 Inflammasome Assay (Promega) was tested but was not sensitive enough to detect a sufficient caspase 1 activity level in human PBMCs isolated from bone marrow or the lymph nodes of the mice. We do not think that the assay is a problem but we rather missed an early time point that would have been the best for T cells since we focused our analysis on the primary end point of viral load, 22 days after HIV infection.

In our experiments, we used lymphocytes isolated from the blood or lymphoid tissues that are probably mostly permissive to HIV infection and give rise to new virions (Cooper et al., 2013; Gougeon et al., 1996). However, in human lymphoid tissues such as tonsil and spleen, HIV enters resting non-permissive cells that represent >95% of the CD4 T cell population according to the group of Doitsh et al. (Doitsh et al., 2010; Eckstein et al., 2001; Moore et al., 2004). As suggested by these authors, these non-permissive cells undergo abortive infection and ultimately die due to an innate immune response launched by the host against cytosolic viral DNA culminating in a caspase 1-dependent pyroptosis (Doitsh et al., 2014; Monroe et al., 2014). Doitsch et al. have also used fresh lymph nodes from subjects infected with R5-tropic HIV, where caspase-1 and IL-1β are nested in the paracortical zone rich in resting CD4 T cells using immunohistochemistry. In our model of humanized mice using busulfan treatment, we observed an improved development of lymphoid structures such as lymph nodes, however, at 23 days after HIV-1 infection, the lymph nodes of mice contain only around 40% of human CD4 + T cells (see Author response image 2). In this lymphoid environment, the Caspase 1 inhibitor should not inhibit productive HIV-1 infection but should block pyroptosis of abortively infected CD4 T cells (Doitsh et al., 2014) mediated by cell-to-cell spread culminating in the pyroptotic death on nonpermissive CD4 T cells. Most of the pathogenic effects of HIV-1 were not attributed to killing of CD4 T cells by circulating free virions but by cell-to-cell contact (Galloway N LK, Cell report 2015). Although several mechanisms of HIV-1 mediated pathogenesis proposed in humans can occur in hematopoietic stem cells-transplanted humanized mice, the immune system poorly supports adaptive immune responses due to the lack of HLA expression in the mouse thymus. At first, there is no evidence yet that such microenvironment with abortive infected CD4 T cells can be reproduced in the lymph nodes or the spleen of humanized mice. However, 22 days after HIV infection, we observed caspase-1 activity only in CD11c^+^ and CD14 ^+^ cells but not in CD4^+^ or CD8 ^+^ T cells. It is important to note, that only the group of Doitsh et al., using a HLAC microenvironment, was able to demonstrate these results. These results were not reproduced by other groups. It is tempting to speculate that, similarly, we were not able to reproduce these results, because we are not using the same type of culture with fresh isolated organs preserving the non permissive and dying CD4 T cells, and therefore we did not catch caspase 1 induced pyroptosis in T cells isolated from the humanized mice.

In addition, a recent study in humanized mice, published in 2021 after the realisation of our experiments suggest that cell death of both naïve and memory CD4 T cells is induced in the spleen at 3 days after the HIV-1 challenge and before the significant loss in peripheral blood (Substantial induction of non-apoptotic CD4 T-cell death during the early phase of HIV-1 infection in a humanized mouse model, Kazutaka Terahara et al., Microbes and infection 2021). The authors used the FLICA technique, and even though dead cells proceeded more degradation of HIV components and direct infection induced cell death, the authors observed that, within live CD4 T cells, Caspase 1 positive cells had more viral reverse-transcripts, suggesting Caspase 1 as an early marker for HIV infection. Furthermore, since the percentages of Caspase 1 positive cells within live CD4 T cells did not significantly increase following HIV challenge (Supplementary Figure 2), the authors suggest that live caspase 1 cells might rapidly die. The authors observed that most dead CD4 T cells were HIV negative in vivo in humanized mice. While these HIV-negative dead cells might have contained cells in which HIV components were degraded, the other cells might have undergone bystander cell death, which is induced when neighboring cells undergo inflammatory cell death, such as pyroptosis and necroptosis, and release proinflammatory mediators that stimulate more cells to die. Therefore, we probably missed the early caspase 1 activity in T cells and just caught the caspase 1 activity in CD11c^+^ and CD14 ^+^ cells that occurred probably later in time in the spleen.

In conclusion, we can assume that caspase 1 positive T cells are dying fast, and that we were not able to detect them, since the number of human T cells in humanized mice is also limited. In our analysis, we used a live dead staining to discriminate between live and dead cells based on membrane integrity. The Live-dead positive cells showed some FLICA staining, as mentionned in Figures 8, 9, and figure 9—figure supplement 1, and VX-765 diminished significantly the number of FLICA positive cells in splenic CD11c^+^ and CD14^+^ cells but not in T cells, where ony residual FLICA staining seems to occur. We analysed the number of FLICA positive cells among the “debris” from the spleen in the flow scatters of the experiment of Figure 8 and Figure 9 for each mouse and identified a level of Caspase 1 activity in the cellular debris of the samples in advanced status of death or necrosis. We observed a mean of 11.6 ±0.8 % ( ± SD) of debris positive for FLICA in Control mice versus 5.1 ± 0.7% (p< 0.05) in VX-765 treated mice, as shown in Author response image 4 for two representative mice. These data indicated that a higher number of Caspase 1 positive cells were found in the cellular debris of HIV-infected huNSG mice as compared to VX-765 treated mice, and could be due to some specific staining to T cells in advanced stage of death. Therefore, we globally assume that we could have missed an early time-point to demonstrate the effects of the caspase-1 inhibitor on dying or pyroptic T cells. However, at 22 days after HIV infection, caspase 1 activity was shown to be mainly decreased by the caspase-1 inhibitor in splenic CD11c^+^ and CD14 ^+^ cells.

**Author response image 4. sa2fig4:** Representative dot plots of FLICA versus LIVE-DEAD staining among debris in splenocytes from vehicle treated (left panel) and VX-765 (right panel) HIV-1 infected huNSG mice.

We added a sentence in the discussion part to clarify that we probably missed pyroptosis of T cells that occurred early in HIV infection as proposed by Kazutaka Terahara et al.

8. The impact on CD4 T cell preservation seems minimal. While there is a statistically significant difference in the blood, it appears to be very slight. Furthermore, this difference does not extend to the spleen. This should be acknowledged.

We agree with the reviewer that the impact of VX-765 is minimal, in contrast to the results of Doitsh et al. This was added in the results and the discussion parts.

It was shown that caspase 1 inhibitor did not inhibit productive HIV-1 infection but block pyroptosis of abortively infected CD4 T cells (Doitsh et al., 2014). In our humanized mice model, although we probably missed the CD4^+^ T cells pyroptosis at earlier time points, the effect of the caspase 1 inhibitor in CD4^+^ T cells depletion was also limited. The fact that the effect of VX-765 is weak together with an early inititiation of pyroptosis that was not sustainable at 22 days after HIV infection, could indicate that the humanized mice model failed to recapitulate a microenvironment leading to T cells pyroptosis. Specifically, cell-to-cell spread of HIV-1 was shown to be required to deplete non-permissive lymphoid CD4 T cells via caspase 1-dependent pyroptosis. This phenomenon can be reduced in an environment where human and murine cells co-exist and only 40% of human T cells are present.

It is important to add again that only one research group (Doitsh et al., 2014) described caspase 1-mediated pyroptosis triggered by abortive viral infection in lymphoid tissues following HIV infection using HLACs. They also demonstrate that the mode of HIV-1 spread determines the outcome form of cell death. The productively infected cell ultimately dies by apoptosis, while the bystander resting cell dies by pyroptosis. Since the killing of CD4^+^ T cells is not attributed to circulating free virions but to infected cells residing in lymphoid tissues that mediate cell-to-cell spread of the virus linked in a single pathogenic cascade, we can assume that the balance between productive and abortive infections could be disturbed in our humanized mice model and did not ensure a rapid depletion of CD4^+^ T cells through pyroptosis. Productively infected cells are obligatorily required to transmit the virus across the virological synapse formed with resting CD4 T cells, and they are decreasing with time in our model, probably faster than in human.

Taken together, all these arguments can justify that fewer bystander resting cells are present in the cells isolated from our mice (as compared to the HDAC model of Doitsh et al) and explain a lower effect of VX-765 CD4^+^ T cell depletion. This was added in the discussion part.

9. The Y axes in figure 6 panels A, B, and C should all be shown in log scale, with no split axes and should all be equivalent to each other as is standard for displaying plasma viral loads in HIV studies.

The changes of the Y axes in log scale, with no split, have been done for Figure 6 panels A, B and C.

10. There is much discussion of the importance of limiting the HIV reservoir and doing so in combination with early-initiated antiretroviral therapy. This discussion should be reduced, as this manuscript does not follow animals beyond day 22 post-infection and does not include animals receiving ART.

We agree with the reviewer and we removed the discussion on HIV reservoirs.

Reviewer #2 (Recommendations for the authors):1. The Authors explained well in the introduction the elimination of HIV reservoirs as a key factor for HIV cure. Also, the Authors reported anti-caspase 1 inhibitor VX-765 reduces HIV reservoirs. However, in the study the Authors did not quantify the different HIV reservoirs (For example Central memory and effector memory CD4 T cells) and the effect of VX-765 on the population of these HIV reservoirs. It will be helpful if the authors could illustrate if VX-765 reduces the population of HIV reservoirs

In this work, we wanted to primarly assess the different effects of caspase-1 inhibition on HIV-1 infection. Our model is a model of HIV-1 infection and not a model of HIV-1 latency by treating animals with cART thereby preserving CD4^+^ T cell depletion. Since CD4^+^ T cells are depleted overtime by HIV-1 infection in the current mice model, we measured the effet of VX-765 only in total HIV-1 DNA isolated from total human CD45+ cells to have enough DNA for quantification. We have previously developed an HIV-1 latency model in humanized NSG mice (P Adams et al., iScience 2021) to investigate the early establishment of the viral reservoir with cART. This model is more appropriate to evaluate the effects of caspase inhibition associated with early cART on viral reservoirs and could better decipher the effects of caspase inhibition in the different memory subsets of T cells. As suggested by reviewer 1, we have decreased the size of the discussion on viral reservoirs, and would suggest to evaluate the effects of inflammasomme inhibitors in central memory or effector memory CD4^+^ T cells in a future work with the latency model of HIV-1 infection.

2. The authors nicely used a negative control for the anti-caspase 1 inhibitor VX-765 during the study. However, there was no positive control. The design of the experiment can be improved by including a positive control like anti-inflammatory drug.

In a previous work, we have published the effect of minocycline, an immunomodulator, in our humnaized mice model, that attenuates HIV-infection and chronic immune activation (M Singh et al., Immunology. 2014 Aug; 142(4): 562–572. Minocycline attenuates HIV-1 infection and suppresses chronic immune activation in humanized NOD/LtsZ-scidIL-2Rγnull mice). We showed that treatment of minocycline at a dose of 100 mg/kg/day for 60 days led to significantly lower viral load and improved T-cell count. Expression of cellular activation (CD38, HLA-DR, CD69 and CCR5) and exhaustion markers (PD-1 and CTLA-4) was significantly lower in the treated group of mice. We knew that the humanized NSG mouse model was able to recapitulate the HIV-induced chronic immune activation observed in infected humans. Since we observed at first a nice up-regulation of the genes involved in inflammasome activation, we did not choose therefore to use a concommittent positive control. Instead, we measured the expression of certain activation markers in CD4^+^ and CD8 + T cells (CD38, CD69, PD-1, HLA-DR) in control and VX-765 treated groups but we did not find any significant differences between the two groups, only a trend displayed in Author response image 5 when the expression of the tree markers was combined.

**Author response image 5. sa2fig5:** Combined expression of CD38, HLA-DR and PD-1 in splenocytes from vehicle-treated and VX-765-treated (right panel) HIV-1 infected huNSG mice.

3. The authors nicely illustrated the expression of genes associated with inflammasome activation in HIV infection. However, this was not done after administration of the anti-caspase 1 inhibitor VX-765. It will make more sense to also illustrate the changes in the transcriptomic profile of markers of inflammasome activation after treatment with VX-765.

Using left-over RNA extracted from human CD45+ cells isolated from the spleen of 5 mice in the Vehicle group and 5 mice in the VX-765 group, at 11 and 22 days after HIV-infection, we measured and normalized the expression of AIM2, IFI16, ASC, and CASP-1mRNA. Treatment with VX-765 reduced significantly their expression at both time points. The results are now added in the manuscript in the Figure 4—figure supplement 1E, F, G, H.

4. Studies have revealed that bystander CD4 T pyroptosis account for about 95% depletion of CD4 T cells in HIV infection https://doi.org/10.1016/j.cell.2010.11.001. However, in this study it was shown that there was more inflammasome activation in infected than uninfected CD4 T cells (Figure 2). Could the Authors expand more of this?

We agree with the reviewer and we think that this remark is very important.

As explained for Reviewer n°1, a recent study in humanized mice, published in 2021 after the realisation of our experiments suggest that cell death of both naïve and memory CD4 T cells is induced in the spleen at 3 days after the HIV-1 challenge and before the significant loss in peripheral blood (Substantial induction of non-apoptotic CD4 T-cell death during the early phase of HIV-1 infection in a humanized mouse model, Kazutaka Terahara et al., Microbes and infection 2021). The authors used the FLICA technique, and even though dead cells proceeded more degradation of HIV components and direct infection induced cell death, the authors observed that, within live CD4 T cells, Caspase 1 positive cells had more viral reverse-transcripts, confirming our data and suggesting that bystander CD4 T pyroptois might happen early during HIV infection. Furthermore, since the percentages of Caspase 1 positive cells within live CD4 T cells did not significantly increase following HIV challenge in this work, the authors suggested that live caspase 1 cells might rapidly die. The authors observed that most dead CD4 T cells were HIV negative in vivo in humanized mice. While these HIV-negative dead cells might have contained cells in which HIV components were degraded, the other cells might have undergone bystander cell death, which is induced when neighboring cells undergo inflammatory cell death, such as pyroptosis and necroptosis, and release proinflammatory mediators that stimulate more cells to die. Therefore, these resuts confirmed that inflamasomme activation occured early in live infected CD4^+^ T cells in our humanized mice model and is then pursued in infected CD11c^+^ and CD14^+^ cells up to 24 days after HIV infection in infected animals. The low caspase 1 activity would indicate that we were not able to recapitulate the bystander effect in humanized mice. We would like also emphasize that the bystander hypothesis was never reproduced so far by other groups of research, other than the group of Doitsh et al. This is now clearly indicated in the discussion.